# Zebrafish macrophage developmental arrest underlies depletion of microglia and reveals Csf1r-independent metaphocytes

Laura E Kuil[1], Nynke Oosterhof[1†], Giuliano Ferrero[2†], Tereza Mikulášová[3], Martina Hason[3], Jordy Dekker[1], Mireia Rovira[2], Herma C van der Linde[1], Paulina MH van Strien[4], Emma de Pater[4], Gerben Schaaf[1], Erik MJ Bindels[4], Valerie Wittamer[2,5‡*], Tjakko J van Ham[1‡*]

[1]Department of Clinical Genetics, Erasmus University Medical Center Rotterdam, Rotterdam, Netherlands; [2]Institut de Recherche Interdisciplinaire en Biologie Humaine et Moléculaire (IRIBHM), Université Libre de Bruxelles (ULB), Brussels, Belgium; [3]Laboratory of Cell Differentiation, Institute of Molecular Genetics of the Czech Academy of Sciences, Prague, Czech Republic; [4]Department of Hematology, Erasmus University Medical Center, Rotterdam, Netherlands; [5]WELBIO, ULB, Brussels, Belgium

**\*For correspondence:**
vwittame@ulb.ac.be (VW);
t.vanham@erasmusmc.nl (TJH)

[†]These authors also contributed equally to this work
[‡]These authors also contributed equally to this work

**Competing interests:** The authors declare that no competing interests exist.

**Abstract** Macrophages derive from multiple sources of hematopoietic progenitors. Most macrophages require colony-stimulating factor 1 receptor (CSF1R), but some macrophages persist in the absence of CSF1R. Here, we analyzed *mpeg1*:GFP–expressing macrophages in *csf1r*-deficient zebrafish and report that embryonic macrophages emerge followed by their developmental arrest. In larvae, *mpeg1+* cell numbers then increased showing two distinct types in the skin: branched, putative Langerhans cells, and amoeboid cells. In contrast, although numbers also increased in *csf1r*-mutants, exclusively amoeboid *mpeg1+* cells were present, which we showed by genetic lineage tracing to have a non-hematopoietic origin. They expressed macrophage-associated genes, but also showed decreased phagocytic gene expression and increased epithelial-associated gene expression, characteristic of metaphocytes, recently discovered ectoderm-derived cells. We further demonstrated that juvenile *csf1r*-deficient zebrafish exhibit systemic macrophage depletion. Thus, *csf1r* deficiency disrupts embryonic to adult macrophage development. Zebrafish deficient for *csf1r* are viable and permit analyzing the consequences of macrophage loss throughout life.

## Introduction

Tissue resident macrophages (TRMs) are phagocytic immune cells that also contribute to organogenesis and tissue homeostasis. Therefore, perturbations in TRM production or activity can have detrimental consequences ranging from abnormal organ development to neurodegeneration and cancer (*Cassetta and Pollard, 2018*; *Mass et al., 2017*; *Yang et al., 2018*; *Zarif et al., 2014*). In vertebrates, including mammals, birds, and fishes, TRMs derive from successive waves of hematopoiesis that initiate early during development reviewed in: *McGrath et al. (2015)*. The initial two embryonic waves give rise to primitive macrophages, born in the embryonic yolk sac in mammals and birds or the rostral blood island (RBI) in fishes, and erythro-myeloid precursors (EMPs), which also originate in the yolk sac and expand in the fetal liver of mammals or emerge from the posterior blood island (PBI) of fishes. A third embryonic wave that generates definitive hematopoietic stem cells (HSCs)

**eLife digest** Immune cells called macrophages are found in all organs in the body. These cells are highly effective at eating and digesting large particles including dead cells and debris, and microorganisms such as bacteria. Macrophages are also instrumental in shaping developing organs and repairing tissues during life.

Macrophages were, until recently, thought to be constantly replenished from cells circulating in the bloodstream. However, it turns out that separate populations of macrophages become established in most tissues during embryonic development and are maintained throughout life without further input.

Previous studies of zebrafish, rodents and humans have shown that, when a gene called *CSF1R* is non-functional, macrophages are absent from many organs including the brain. However, some tissue-specific macrophages still persist, and it was not clear why these cells do not rely on the *CSF1R* gene while others do.

Kuil et al. set out to decipher the precise requirement for the *CSF1R* gene in macrophage development in living zebrafish. The experiments used zebrafish that make a green fluorescent protein in their macrophages. As these fish are transparent, this meant that Kuil et al. could observe the cells within the living fish and isolate them to determine which genes are switched on and off. This approach revealed that zebrafish with a mutated version of the *CSF1R* gene make macrophages as embryos but that these cells then fail to multiply and migrate into the developing organs. This results in fewer macrophages in the zebrafish's tissues, and an absence of these cells in the brain.

Kuil et al. went on to show that new macrophages did emerge in zebrafish that were about two to three weeks old. However, unexpectedly, these new cells were not regular macrophages. Instead, they were a new recently identified cell-type called metaphocytes, which share similarities with macrophages but have a completely different origin, move faster and do not eat particles.

Zebrafish lacking the *CSF1R* gene thus lose nearly all their macrophages but retain metaphocytes. These macrophage-free mutant zebrafish constitute an unprecedented tool for further studies looking to discriminate the different roles of macrophages and metaphocytes.

---

begins in the aorta-gonad-mesonephros (AGM) region, where HSCs bud from the hemogenic endothelium (*Bertrand et al., 2010*; *Boisset et al., 2010*; *Kissa and Herbomel, 2010*). In zebrafish, newly born hematopoietic stem cells (HSCs) migrate to the caudal hematopoietic tissue (CHT), and later seed hematopoietic organs such as the kidney marrow, which is equivalent to the bone marrow in mammals (*Henninger et al., 2017*; *Murayama et al., 2006*). Most TRM populations are established by the end of fetal life and are subsequently maintained through the proliferation of local progenitors or through the partial contribution of bone marrow-derived cells (*Liu et al., 2019*).

During their colonization of the embryo, macrophages acquire distinct properties adapted to their microenvironment and allowing them to execute tissue niche-specific functions (*Bennett and Bennett, 2020*; *Gosselin et al., 2014*; *Gosselin et al., 2017*; *Lavin et al., 2014*; *Matcovitch-Natan et al., 2016*). The ontogeny of TRMs within a specific organ is heterogeneous and thought to be determined by the availability of the niche and accessibility of the host tissue reviewed in: *Guilliams et al. (2020)*. The microenvironment has a major role in determining TRM phenotype and function, largely regardless of ontogeny, but giving rise to heterogeneous populations of cells (*Lavin et al., 2014*; *Shemer et al., 2018*; *van de Laar et al., 2016*).

Colony stimulating factor 1 receptor (CSF1R) is an evolutionarily conserved regulator of macrophage development, directly inducing DNA and protein synthesis as well as proliferation upon ligand binding (*Hume et al., 2016*; *Tushinski and Stanley, 1985*). Recessive and dominant mutations in *CSF1R* can cause severe brain disease (*Konno et al., 2018a*; *Konno et al., 2018b*; *Oosterhof et al., 2019*; *Rademakers et al., 2012*), associated with lower microglia density (*Oosterhof et al., 2018*), but whether such mutations affect other myeloid cells, and how, remains unknown. Recently, patients carrying homozygous mutations in *CSF1R* and presenting with both leukodystrophy and osteopetrosis, phenotypes attributed to an absence of TRMs in the brain and bone, have been described (*Oosterhof et al., 2019*). In mice and rats, the absence of CSF1R results in a complete lack of microglia, Langerhans cells (LCs), and osteoclasts, while other subsets of TRMs

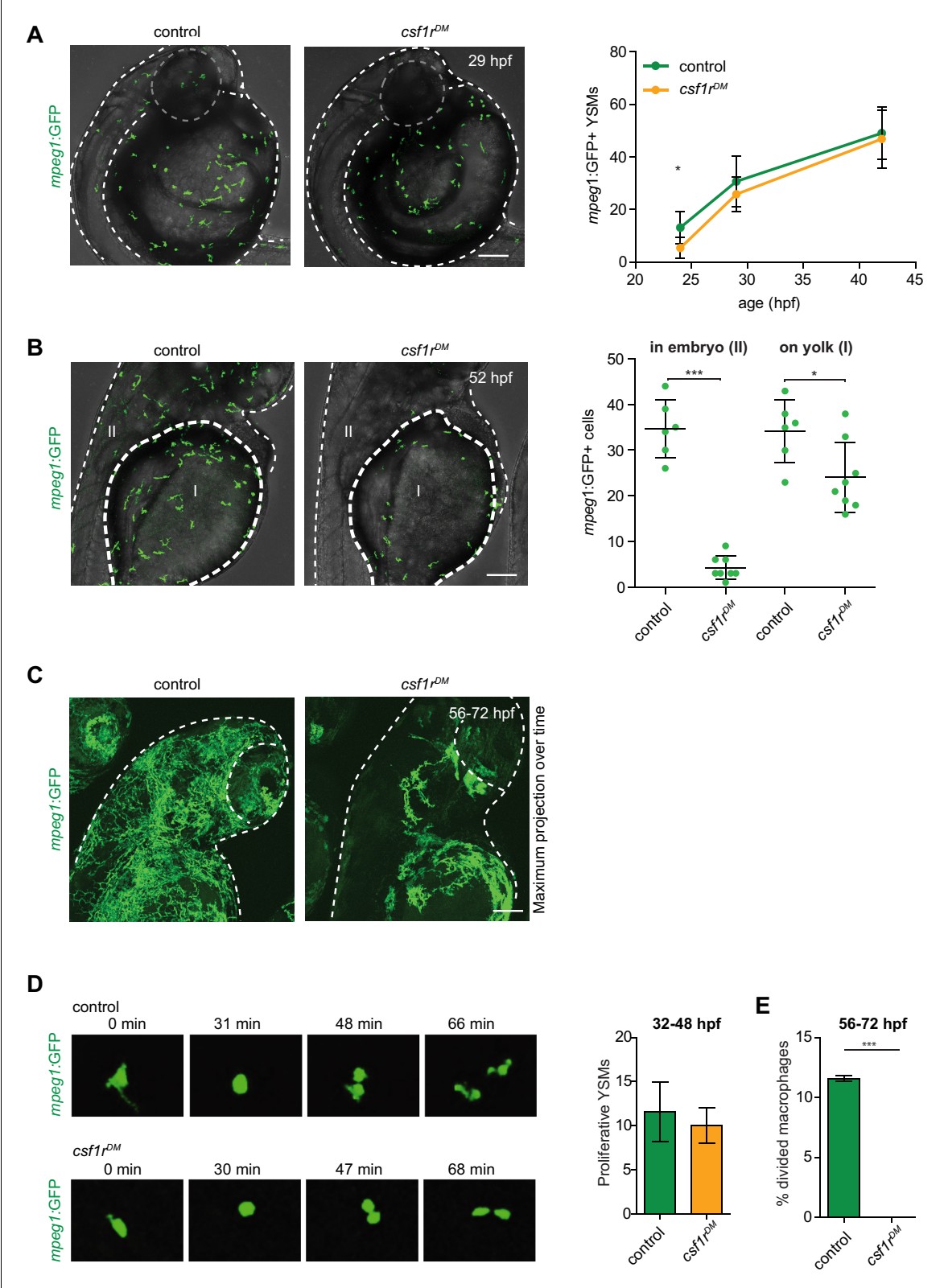

**Figure 1.** *mpeg1+* primitive macrophages on the yolk in control and *csf1r^{DM}* larvae. (**A**) Representative images of *mpeg1+* macrophages located on the yolk (29 hpf) and quantification of *mpeg1+* cell numbers over time. (**B**) Representative images of *mpeg1+* positive primitive macrophages at 52 hpf. The dotted line indicates the border between the yolk (I) and the embryonic tissue (II). Quantification of *mpeg1+* macrophages that colonized the tissue (II) and primitive macrophages located on the yolk (I). (**C**) Representative maximum projection of long term time lapse imaging of control and *csf1r^{DM}*

*Figure 1 continued on next page*

*Figure 1 continued*

larvae showing migratory trajectories of *mpeg1+* macrophages. (D) Snap shots from dividing *mpeg1+* primitive macrophages in control and *csf1r*$^{DM}$ larvae (~36 hpf) and quantification of proliferative primitive macrophages during 16 hr time lapse imaging (~32 hpf – 48 hpf) (control n = 5, *csf1r*$^{DM}$ n = 3). (E) Quantification of fraction proliferative embryonic macrophages during 16 hr time lapse imaging (~56 hpf – 72 hpf) in control and *csf1r*$^{DM}$ larvae (n = 3 per group). Scale bars represent 100 µM. Error bars represent standard deviation. Statistical significance is calculated using one-way ANOVA with Dunnett's multiple comparison test or Student's *t*-tests *<0.05 **<0.01 ***<0.001. *mpeg1+* cells were quantified on one side of the embryo (right side). Each dot represents one fish.

The online version of this article includes the following figure supplement(s) for figure 1:

**Figure supplement 1.** *Mpeg1+* cells can be detected in the tail region of control and *csf1r*$^{DM}$ larvae.

are affected to varying degrees (*Cecchini et al., 1994*; *Dai et al., 2002*; *Erblich et al., 2011*; *Ginhoux et al., 2010*; *Oosterhof et al., 2018*; *Pridans et al., 2018*). It is unknown whether CSF1R is required for the development of early, embryonic TRM precursors and it remains elusive as to why only specific TRM populations are lacking in the absence of *Csf1r*. Furthermore, it is unclear whether macrophages that persist in *Csf1r*-deficient mice and rats have a normal macrophage phenotype. Detailed analysis of the *Csf1r* mutant phenotypes could therefore contribute to the identification of specific and universal features of organism-wide macrophage development. In addition, it is important to understand the systemic effects of CSF1R inhibition on macrophages, as inhibition of CSF1R is a clinical strategy for the intentional depletion of macrophages in various disease contexts, including Alzheimer's disease, brain injury and cancer (*Edwards et al., 2019*; *Lloyd et al., 2019*; *Tap et al., 2015*; *Webb et al., 2018*).

Zebrafish are particularly suitable to study immune cell development in vivo as they develop ex utero, are genetically tractable, and are transparent during early development (*Ellett and Lieschke, 2010*; *Gore et al., 2018*). We used our previously generated zebrafish line that is deficient for both *csf1ra* and *csf1rb* paralogs (*csf1r*$^{DM}$), since the phenotypes of these fish, such as osteopetrosis and a lack of microglia, resemble those observed in mice, rats and humans (*Caetano-Lopes et al., 2020*; *Chatani et al., 2011*; *Dai et al., 2002*; *Guo et al., 2019*; *Meireles et al., 2014*; *Oosterhof et al., 2019*; *Oosterhof et al., 2018*; *Pridans et al., 2018*). The strong homology of basic developmental cellular processes has proven this model as indispensable for the identification of novel basic features of immune cell development and function (*Barros-Becker et al., 2017*; *Bertrand et al., 2010*; *Espín-Palazón et al., 2014*; *Kissa and Herbomel, 2010*; *Madigan et al., 2017*; *Tamplin et al., 2015*; *Tyrkalska et al., 2019*).

Here, we aimed to determine how and when loss of Csf1r affects macrophage development. We found that primitive myelopoiesis is initially *csf1r*-independent, although *csf1r*$^{DM}$ embryonic macrophages subsequently ceased to divide and failed to colonize embryonic tissues. Surprisingly, a detailed examination of *csf1r*$^{DM}$ larval zebrafish revealed another wave of *mpeg1+* cells in the skin from 15 days of development onwards, but these cells lacked the branched morphology typical of Langerhans cells (*He et al., 2018*). Using fate mapping and gene expression profiling, we identified *csf1r*$^{DM}$ *mpeg1+* cells as metaphocytes, a population of ectoderm-derived macrophage-like cells recently reported in zebrafish (*Alemany et al., 2018*; *Lin et al., 2019*). Extending our analyses, we further demonstrated that adult *csf1r*$^{DM}$ fish exhibit a global defect in macrophage generation. In conclusion, our study highlights distinct requirements for Csf1r during macrophage generation and metaphocyte ontogeny, resolving part of the presumed macrophage heterogeneity and their sensitivity to loss of Csf1r.

## Results

### Zebrafish embryonic macrophages are formed independently of *csf1r* but display migration and proliferation defects

To determine whether the earliest embryonic macrophages, called primitive macrophages, are still formed in the absence of Csf1r signaling, we analyzed *csf1r*$^{DM}$ zebrafish embryos carrying the macrophage transgenic reporter *mpeg1:GFP* (*Ellett et al., 2011*; *Oosterhof et al., 2018*). Zebrafish primitive macrophages are born in the rostral blood island on the yolk and can be detected by *mpeg1:GFP* expression from 22 hr post fertilization (hpf) as they migrate on the yolk ball—

equivalent to the mammalian yolk sac—and progressively invade peripheral tissues (*Herbomel et al., 1999*; *Herbomel et al., 2001*). These constitute the main macrophage population during the first 5 days of development (*Wu et al., 2018*). Indeed, in vivo imaging of GFP-expressing macrophages in control embryos showed that, at 24 hpf, ~13 *mpeg1+* primitive macrophages were present on the yolk, increasing to ~49 cells at 42 hpf (*Figure 1A*, *Video 1*; *Ellett et al., 2011*). In *csf1r^{DM}* embryos, even though primitive macrophage numbers were slightly lower at 24 hpf (~5 *mpeg1+* cells), macrophage numbers did not significantly differ from controls at 42 hpf (~46 *mpeg1* + cells) (*Figure 1A*). This indicates that Csf1r is dispensable for the emergence of primitive macrophages.

We next investigated whether embryonic macrophages in *csf1r^{DM}* animals retained the ability to invade peripheral tissues. At 52 hpf, 50% of *mpeg1+* cells had exited the yolk epithelium in controls and were observed in the periphery (*Figure 1B*). In contrast, only 15% of all macrophages were found outside of the yolk in *csf1r^{DM}* embryos. At this stage, macrophage numbers were significantly lower in *csf1r^{DM}* larvae than controls (*Figure 1B*). Migration trajectories of embryonic macrophages into the embryonic tissues, as shown by maximum intensity projections of images acquired over 16 hr, were more widespread in controls than *csf1r^{DM}* and covered the entire embryo (*Figure 1C*, *Video 2*). Thus, although the generation of embryonic macrophages appeared independent of *csf1r*, after two days of development macrophage failed to expand in the *csf1r* mutants and their migration was reduced, suggesting functional deficits caused by the loss of Csf1r.

We hypothesized that the reduced macrophage numbers in *csf1r* mutants could be explained by a reduction in their proliferative activity. To test this, we performed live imaging on *mpeg1+* cells and quantified cell divisions. Between 32 and 48 hpf, the proliferative rates were not significantly different between control (~12 events) and *csf1r^{DM}* embryos (~10 events) (*Figure 1D*, *Video 1*). However, whereas control macrophages actively proliferated between 56 and 72 hpf (~11% of macrophages divided), *csf1r^{DM}* macrophages did not (none of the macrophages divided) (*Figure 1E*). This indicates that the expansion of primitive macrophages is halted between 48 and 56 hpf. Thus, while the initial proliferation of emerging primitive macrophages occurs independent of *csf1r*, by 48 hpf Csf1r signaling becomes necessary for embryonic macrophage proliferation.

## RNA-sequencing of embryonic macrophages reveals *csf1r*-independent core macrophage differentiation

To explore specific developmental and molecular processes affected by the loss of Csf1r signaling, and to discern a potential effect on proliferation, we performed RNA sequencing on macrophages isolated from 28 and 50 hpf *mpeg1:GFP* embryos using fluorescence-activated cell sorting (FACS). These time points were chosen to study the primitive macrophages soon after their emergence from the RBI (28 hpf) and as they subsequently transition to a tissue colonizing, migratory phenotype (50 hpf) (*Figure 2A*). Principal component analysis (PCA) of the macrophage gene expression data sets showed clustering of triplicate samples based on genotype (component 1) and developmental stage (component 2) (*Figure 2B*). This suggests that, even though gene expression differed between

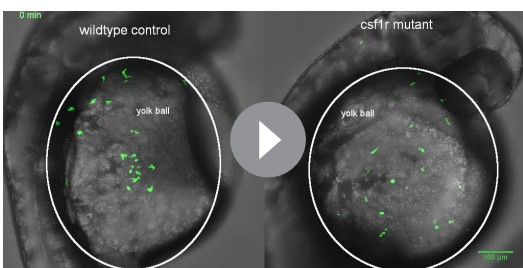

**Video 1.** Time-lapse recording of primitive macrophages on the yolk from 32 to 48 hpf showing frequent proliferative events in both control and *csf1r^{DM}* embryos.

https://elifesciences.org/articles/53403#video1

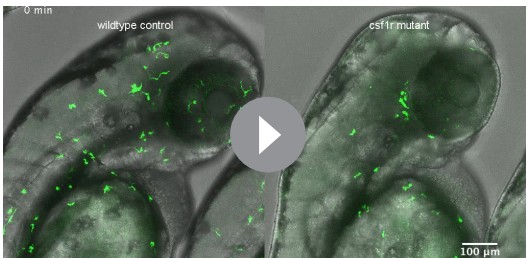

**Video 2.** Time-lapse recording from 56 to 72 hpf, showing the colonization of the embryo by macrophages in control and the migration defect observed in *csf1r^{DM}* embryos.

https://elifesciences.org/articles/53403#video2

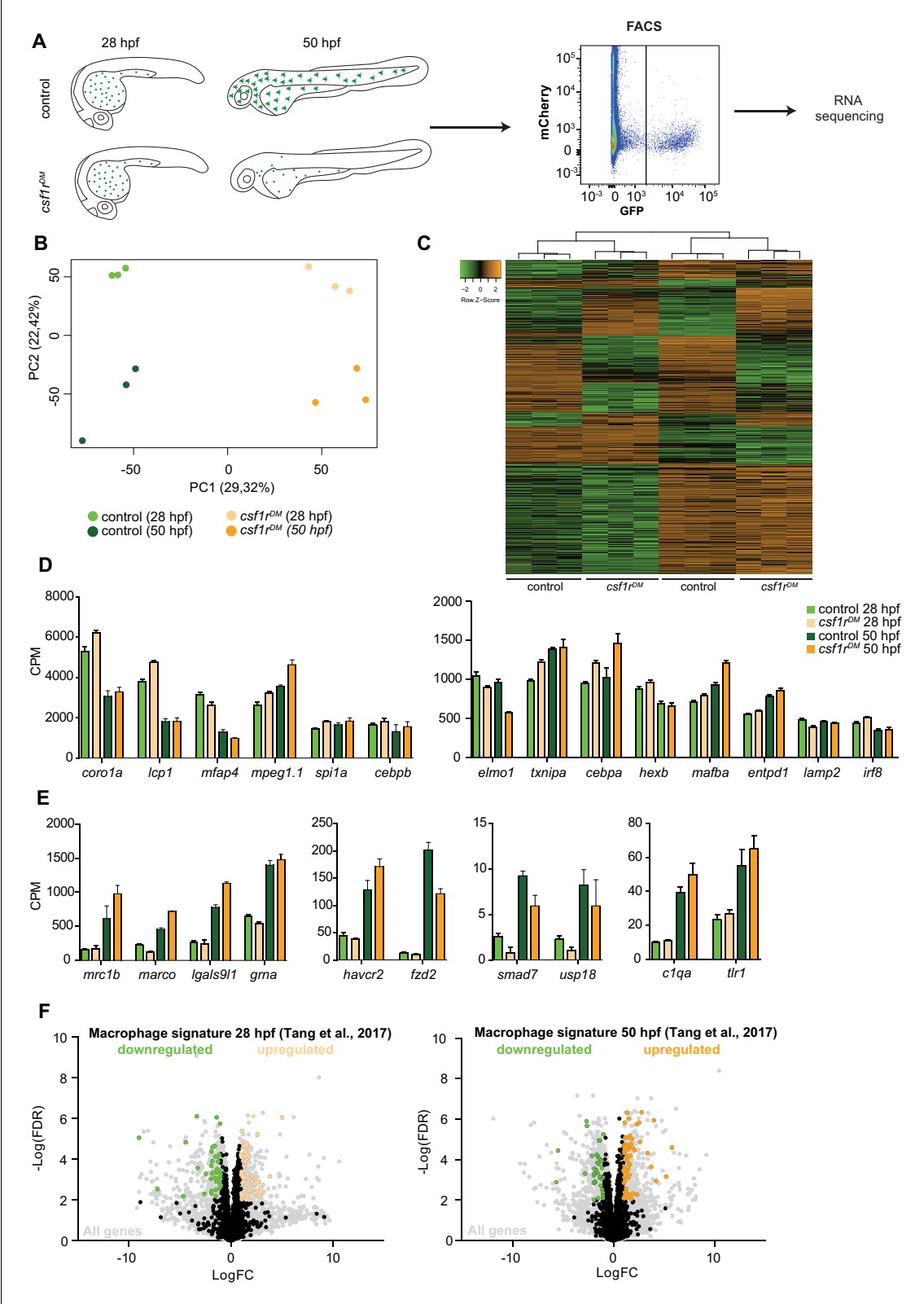

**Figure 2.** RNA sequencing of primitive macrophages at different developmental stages reveals cell cycle arrest in *csf1r*^DM macrophages from 2 dpf onward. (**A**) Schematic representation of the experimental set-up. *mpeg1+* cells were isolated from both control and *csf1r*^DM larvae at 28 hpf and 50 hpf using FACS. These cells were used for RNA sequencing. (**B**) PCA analysis shows clustering of triplicates and segregation on genotype (component 1) and developmental stage (component 2). (**C**) Heat map showing all significantly differentially expressed genes (logFC > |1|; FDR < 0.01). (**D**) Counts per
*Figure 2 continued on next page*

Figure 2 continued

million (CPM) of 'macrophage signature' genes show high, non-differential expression in all groups (logFC > |1|; FDR > 0.01). (E) CPM values of 'macrophage signature' genes induced over time in control and *csf1r^DM* macrophages (logFC > |1|; FDR < 0.01). (F) Volcano plot showing genes expression changes between control and *csf1r^DM* at 28 hpf and 50 hpf respectively. Light grey: all reads, Black/Green/Orange: Macrophage/myeloid signature genes based on data from *Tang et al. (2017)* (*Tang et al., 2017*); Black: non-differentially expressed between controls and *csf1r^DM* macrophages; Green: significantly upregulated in control macrophages; Orange: significantly upregulated in *csf1r^DM* macrophages (logFC > |1|; FDR < 0.01). 4% and 5% of the macrophage genes were significantly differentially expressed between control and *csf1r^DM* macrophages at 28 and 50 hpf respectively.

The online version of this article includes the following figure supplement(s) for figure 2:

**Figure supplement 1.** *csf1r^DM* macrophages and microglia have a proliferation defect.

control and *csf1r^DM* macrophages at both time points, most of the changes that occurred over time in control embryos also occurred in *csf1r^DM* embryos (*Figure 2B,C*). To determine macrophage identity we analyzed the expression of genes highly expressed in macrophages, including genes used in zebrafish as macrophage markers (e.g. *csf1ra, mfap4*), chemokine and pathogen recognition receptors (e.g. *marco, mrc1, tlr1*), and myeloid transcription factors (e.g. *irf8, spi1a, cebpb*), but we did not observe major differences between genotypes (*Figure 2D–E*). Also, when we compared our gene expression profiles with a zebrafish macrophage expression profile determined by single cell RNA-seq (*Tang et al., 2017*), only ~5% of the reported 2031 macrophage-specific genes were differentially expressed in *csf1r^DM* macrophages, suggesting Csf1r-independent expression of the majority of these macrophage-expressed genes (*Figure 2F*). Together, this shows that *csf1r*-deficient embryonic macrophages display a core gene expression profile similar to that seen in controls.

## Impaired proliferation of embryonic *csf1r^DM* macrophages is reflected in their transcriptome and proliferation is not restored in microglia

The nature of the differences in gene expression profiles between control and *csf1r^DM* macrophages was studied by gene set enrichment analysis (GSEA). GSEA revealed that, at both time points, *csf1r^DM* macrophages had lower expression of genes associated with RNA metabolism and DNA replication (*Figure 3A*), with transcripts encoding all components of the DNA replication complex being ~2 fold reduced (*Figure 2—figure supplement 1* 3B). In addition, *csf1r^DM* macrophages showed lower expression of genes in GO classes related to cell cycle at 50 hpf (*Figure 3A*, *Figure 2—figure supplement 1*). Thus, at 28 hpf, DNA replication genes were downregulated, followed by a decrease in expression of genes involved in general cell cycle related processes at 50 hpf. Together, and in line with our in vivo findings, these analyses suggest that proliferation is reduced or halted in *csf1r^DM* macrophages from 2 dpf onwards.

Of the three Csf1r ligand genes, both *csf1a* and *csf1b* are expressed at 20 hpf, whereas *il34* is not detectable at that time, barely detectable at 24 hpf, and moderately expressed at 36 hpf (*Figure 2—figure supplement 1C*). Therefore, it is possible that the reduced expression of cell cycle related genes in *csf1r^DM* macrophages could be attributed largely to a lack of interaction between the two Csf1 ligands and Csf1r. Additionally, this suggests that these two ligands likely do not influence the specification of embryonic macrophages at this stage. Previous analyses of macrophage development in *il34^-/-* deficient zebrafish around 30 hpf showed primarily a deficiency in the migration of macrophages across the embryo and into the brain (*Kuil et al., 2019*; *Wu et al., 2018*).

Microglia are the first TRM population present during embryonic development and they are highly proliferative during this time (*Ginhoux et al., 2010*; *Herbomel et al., 2001*; *Xu et al., 2016*). Therefore, we determined whether loss of Csf1r signaling also affects microglial proliferation. Pcna/ L-plastin double immunostaining in control embryos showed that total microglia numbers increase between 2 and 4 dpf. At 2 dpf almost no macrophages in the brain are proliferating, whereas ~20% of the population is at 4 dpf (*Figure 2—figure supplement 1B*). In *csf1r^DM* larvae a few microglia were occasionally present in the brain between 2 and 4 dpf, however none were Pcna+ (*Figure 2— figure supplement 1B*). EdU pulse labeling experiments, marking cells that proliferated between 4 and 5 dpf, showed no EdU+ microglia in *csf1r* mutants, suggesting that *csf1r*-deficient microglia fail to proliferate (*Figure 3C*). Thus, proliferation is impaired in both *csf1r^DM* primitive macrophages and early microglia.

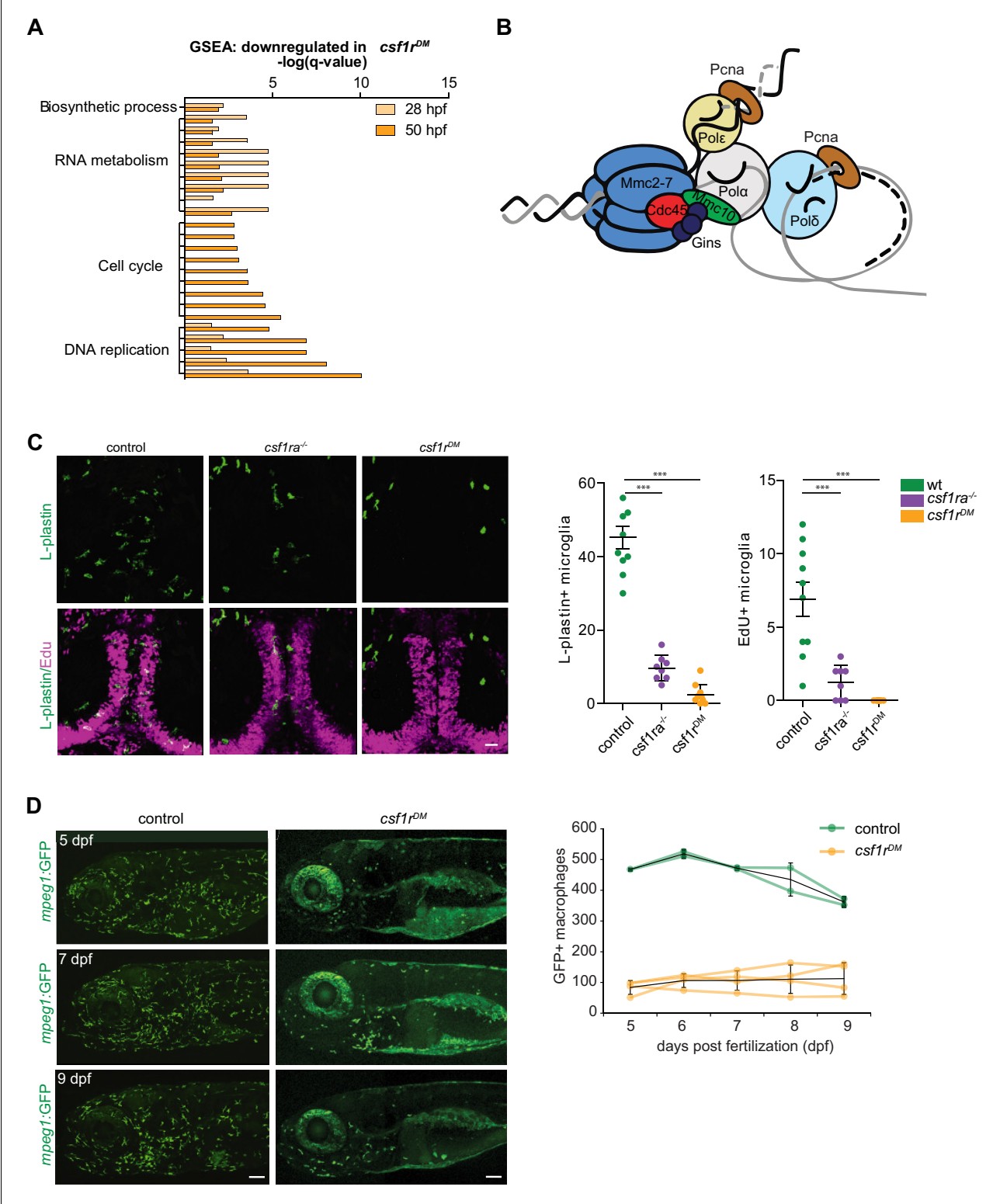

**Figure 3.** Csf1r-deficient tissue resident macrophages (microglia) fail to proliferate. (**A**) Bar graph showing the GO terms associated with enriched genes downregulated in *csf1r^{DM}* macrophages (p<0,05). (**B**) Cartoon representing the vertebrate DNA replication complex, all components were significantly downregulated in *csf1r^{DM}* macrophages. (**C**) Representative images and quantification of L-plastin/Edu double positive microglia at 5 dpf. Scale bar represents 25 µM. (**D**) Representative images, and quantification, of *mpeg1*+ macrophages in the anterior part of 5, 7 and 9 day old zebrafish and quantification of total number of macrophages at the imaged half of the total embryo. *Mpeg1*+ cells were quantified on one side of the embryo

*Figure 3 continued on next page*

*Figure 3 continued*

(right side). Error bars represent standard deviation. Statistical significance is calculated using one-way ANOVA with Dunnett's multiple comparison test *<0,05 **<0,01 ***<0001. Each dot represents one fish.

Next, we assessed the presence of macrophages in developing *csf1r*<sup>DM</sup> animals by in vivo fluorescence imaging of one lateral side of entire, individual larvae on 4 consecutive days, starting at 5 dpf. We visualized ~450 macrophages in control animals, whereas *csf1r*<sup>DM</sup> animals contained >4 fold fewer (~100) (*Figure 3D*). Over the next 4 days, macrophage numbers in both groups remained stable (*Figure 3D*). This suggests that, at this stage, there is neither proliferative expansion of embryonic macrophages nor supply of macrophages from an alternative source, causing macrophage numbers in *csf1r*<sup>DM</sup> larvae to remain much lower than those in controls up to 9 dpf. Together these data indicate that, onwards from the initiation of embryonic tissue colonization, proliferative expansion of macrophages remains halted in *csf1r*<sup>DM</sup> animals.

## *csf1r*<sup>DM</sup> skin lacks highly branched putative Langerhans cells

Given that macrophages are produced by consecutive waves of primitive and definitive myelopoiesis, and that embryonic *csf1r*<sup>DM</sup> macrophages ceased to proliferate, we wondered whether macrophages would be present at later developmental stages in *csf1r*<sup>DM</sup> zebrafish. By live imaging at 20 dpf we detected *mpeg1*+ cells in the skin of control animals, as expected, but also in the skin of *csf1r*<sup>DM</sup> animals (*Figure 4A*). To pinpoint the emergence of these *mpeg1*+ cells we live imaged entire zebrafish unilaterally from 8 until 24 dpf (*Figure 4B*). Between 10 and 13 dpf, control *mpeg1*+ cell numbers increased ~1.6 fold and *csf1r*<sup>DM</sup> *mpeg1*+ cell numbers increased 2.4 fold (*Figure 4B*). From 15 to 17 dpf onwards, *mpeg1*+ cell numbers continued to increase exponentially both in controls and in *csf1r*<sup>DM</sup> fish. As we noticed differences in the size of the zebrafish, as they grew older, both among controls and mutants, we also plotted *mpeg1*+ cell numbers against fish size (*Figure 4B*). Larval zebrafish rapidly grow in size, and their size often correlates better with developmental hallmarks than their age in days (*Parichy et al., 2009*). In larval fish smaller than 5 mm, *mpeg1*+ cell numbers did not increase, whereas in fish that were larger than 5 mm *mpeg1*+ cell numbers correlated almost linearly with size. Taken together, we show that particularly in larvae older than 15 dpf, or over 5 mm in size, *mpeg1*+ cell numbers increase significantly, independent of *csf1r* mutation status.

Despite the overall similar kinetics of *mpeg1*+ cell emergence, we observed major morphological differences in these cells between control and *csf1r*<sup>DM</sup> animals. In the skin of 22 dpf control zebrafish, we found two distinct cell morphologies: those presenting with a branched and mesenchymal cell shape reminiscent of mammalian Langerhans cells, the macrophage population in the epidermis, and those that display a compact, amoeboid morphology with short, thick, primary protrusions (*Figure 4C*). In 22 dpf *csf1r*<sup>DM</sup> fish, only the more amoeboid cell type was present. These persisting amoeboid *mpeg1*+ cells in *csf1r* mutant animals could represent a subtype of macrophages, or skin metaphocytes, a newly identified macrophage-like cell type (*Alemany et al., 2018*; *Lin et al., 2019*).

Metaphocytes are ectoderm-derived cells that display gene expression overlapping partly with macrophages, including *mpeg1*, but with much lower expression of phagocytosis genes; these cells also lack a phagocytic response upon infection or injury (*Alemany et al., 2018*; *Lin et al., 2019*). As metaphocytes have also been reported to migrate faster than skin macrophages and morphologically resemble the *mpeg1*+ cells that remain in *csf1r*<sup>DM</sup> fish, we used time-lapse imaging and showed that, both in controls and in *csf1r*<sup>DM</sup> fish, the smaller, amoeboid *mpeg1*+ cells were highly motile (*Video 3*; *Lin et al., 2019*). In contrast, the branched *mpeg1*+ cells that were found only in controls presented long, continuously extending and retracting protrusions and an evenly spaced distribution, but were largely confined to their location during 3 hr imaging periods. These highly branched macrophages, which were absent in *csf1r*<sup>DM</sup> fish, were located in the skin epidermis and, based on their location, morphology, migration speed, and behavior, may represent the zebrafish counterpart to mammalian Langerhans cells (*Video 3 Lugo-Villarino et al., 2010*). In support of this notion, branched *mpeg1*+ cells were hardly detected in the skin of zebrafish deficient for interleukin-34 (*Figure 4—figure supplement 1A*; 4C), the Csf1r ligand that selectively controls the development of Langerhans cells in mice (*Greter et al., 2012*; *Wang et al., 2012*). In larval zebrafish, *csf1a* and *csf1b*

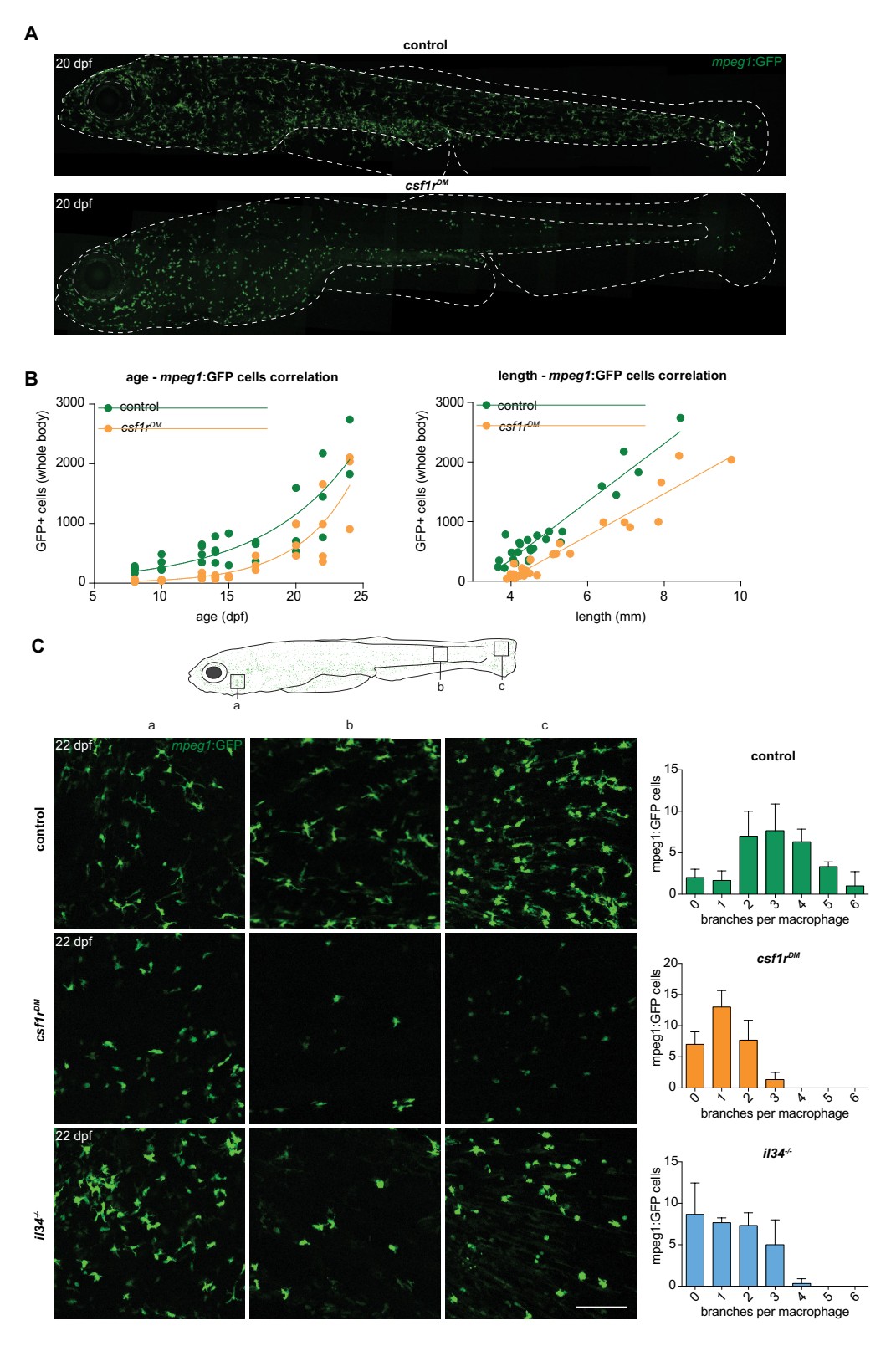

**Figure 4.** Two morphologically distinct populations of *mpeg1+* cells in emerge from 15 dpf in the zebrafish skin. (**A**) Representative images of a control and *csf1r^DM* zebrafish at 20 dpf. Dotted line represents the outline of the fish and its eye. (**B**) Quantification of the total number of *mpeg1+* cells at one unilateral side of the fish at different time points between 8 and 24 dpf. The number of mpeg1+ cells was manually counted from the unilateral side presented in panel **A**. Plot showing the relationship between number of *mpeg1+* cells and fish size. Each dot represents one fish. (**C**) Representative
*Figure 4 continued on next page*

*Figure 4 continued*

images of *mpeg1+* cells in different body regions at 22 dpf showing differences in morphology between controls and *csf1r^DM* or *il34^-/-mpeg1+* cells (n = 3 per group). Error bars represent standard deviation. *Mpeg1+* cells were quantified on one side of the embryo (right side).

The online version of this article includes the following figure supplement(s) for figure 4:

**Figure supplement 1.** Abnormal morphology of *csf1r^DM* and *il34^-/-*, but not *csf1a^-/-b^-/-* larval *mpeg1+* cells in the skin.

expression were detected in skin (*Figure 2—figure supplement 1*), more specifically in interstripe iridophores and hypodermal and fin cells (*Patterson and Parichy, 2013*). Although we found that *il34* was also expressed in adult skin, this expression was about 10-fold lower than that of *csf1a* or *csf1b* (*Figure 2—figure supplement 1D*). However, our in vivo imaging data suggests that the loss of Il34, but not of both Csf1a and Csf1b, affects branched skin macrophages in particular (*Figure 4—figure supplement 1B*).

## Remaining *mpeg1+* cells in *csf1r^DM* skin are metaphocytes

We reasoned that macrophages, and/or possibly Langerhans cells, could be absent in *csf1r^DM* and *il34* mutant skin, and that remaining *mpeg1+* cells may be metaphocytes. Unlike macrophages, metaphocytes are of non-hematopoietic, likely ectodermal origin (*Lin et al., 2019*). We recently proposed that skin macrophages and metaphocytes, based on these different ontogenies, could be discriminated in the adult zebrafish using the Tg(*kdrl*:Cre; *ßactin2*:loxP-STOP-loxP-DsRed) fate-mapping model that labels EMPs, HSCs and their progenies (*Bertrand et al., 2010*; *Ferrero et al., 2018*). Genetic, permanent labeling with DsRed of adult leukocytes, including branched skin macrophages is induced by constitutive expression of Cre recombinase in endothelial cells and hemogenic endothelium (*Bertrand et al., 2010*). As suggested by restricted expression of the metaphocyte marker *cldnh* in *mpeg1*-GFP^+DsRed^- cells, non-hematopoietic metaphocytes lack DsRed labeling (*Ferrero et al., 2020*). The presence or absence of DsRed expression could thus be used to discriminate between metaphocytes (GFP^+DsRed^-) and macrophages (GFP^+DsRed^+). Of note, a possible caveat is that *mpeg1+* primitive macrophages, which derive directly from *kdrl*-negative mesoderm, are also not marked by DsRed in this setting, which could complicate the interpretation of results. However, as we previously documented, there seems to be no contribution from primitive hematopoiesis to *mpeg1*-expressing cells in the adult skin (*Ferrero et al., 2020*). In addition, primitive macrophages appear virtually absent in Csf1r-deficient zebrafish, thus making this approach suitable to address the identity of *mpeg1+* cells in *csf1r^DM* skin. We generated *csf1r*-deficient animals carrying these three transgenes and examined their skin by confocal imaging. In control adult zebrafish skin, populations both of GFP^+DsRed^+ and of GFP^+DsRed^- cells were present, while only GFP^+DsRed^- cells could be detected in *csf1r^DM* animals (*Figure 5A*). This phenotype was further validated by flow cytometry analysis, showing a ~ 90% decrease in the GFP^+DsRed^+ population in *csf1r^DM* zebrafish skin but no change in the frequency of GFP^+DsRed^- cells (*Figure 5B*). Collectively, these results suggest that the generation of skin definitive macrophages is largely Csf1r-dependent and point to metaphocytes as the remaining *mpeg1+* cells in *csf1r^DM* skin.

To further characterize cell identity, we FAC-sorted GFP^+DsRed^+ and GFP^+DsRed^- cells from control fish skin and GFP^+DsRed^- cells from *csf1r^DM* skin and performed bulk RNA sequencing. PCA shows clustering of duplicates and segregation of GFP^+DsRed^- and GFP^+DsRed^+ (PC1) and genotype (PC2) (*Figure 5C*). Consistent with their expected hematopoietic identity, GFP^+DsRed^+ cells expressed the pan-leukocyte marker *ptprc* (*Figure 5—figure supplement 2A*). In contrast, GFP^+DsRed^- cells were negative for this marker. To address whether GFP^+DsRed^- cells overlap with metaphocytes, we selected genes

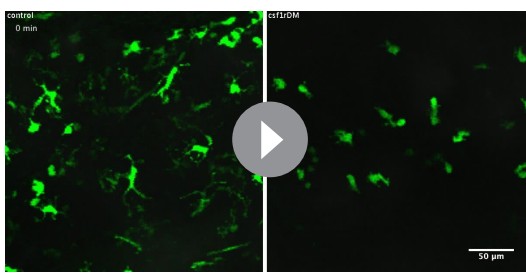

**Video 3.** 3 hr time-lapse recordings of macrophages in the skin showing branched, mesenchymal macrophages and non-branched, amoeboid metaphocytes in control fish and only non-branched, amoeboid metaphocytes in *csf1r^DM* fish.
https://elifesciences.org/articles/53403#video3

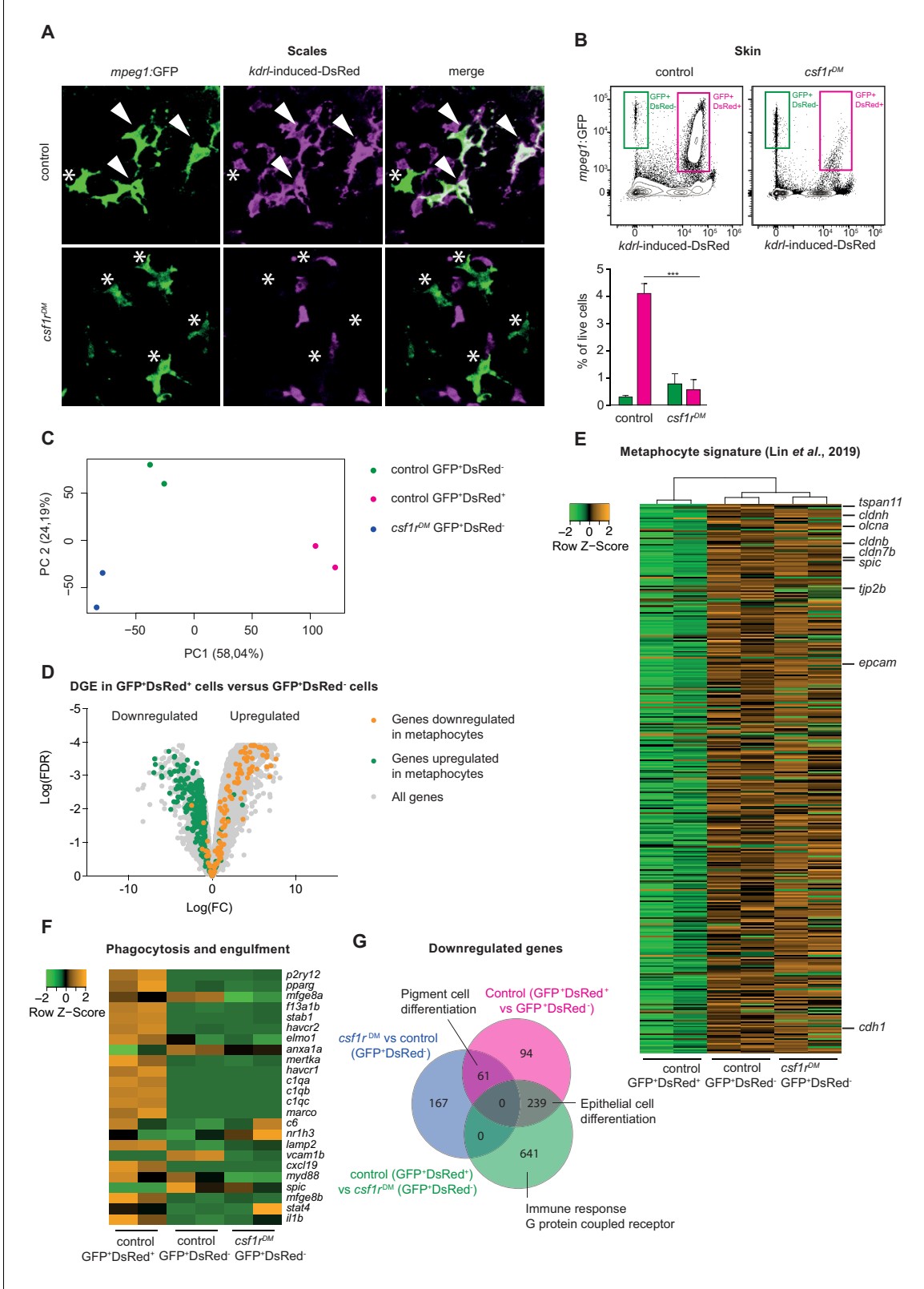

**Figure 5.** Amoeboid *mpeg1+* cells in the zebrafish skin are of non-hematopoietic origin and have a metaphocyte transcriptome. (**A**) Immunofluorescence on manually dissected scales from adult skin of control and *csf1r^DM mpeg1*:EGFP +; *kdrl*-induced-DsRed+ adults (4 mpf). Stars: single-positive (SP) cells; white arrowheads: double-positive (DP) cells. (**B**) FACS analysis on cells from the adult skin (4 mpf, n = 3 per group) and quantification. GFP⁺DsRed⁻=*mpeg1*+ only, GFP⁺DsRed⁺=*mpeg1*+/*kdrl*-induced-DsRed+. (**C**) PCA analysis showing segregatin based on cell type (PC1)

*Figure 5 continued on next page*

Figure 5 continued

and genotype (PC2). (D) Volcano plot showing gene expression changes between control GFP⁺DsRed⁺ versus GFP⁺DsRed⁻ cells. Light grey: DGE of all genes, Green: DGE of genes enriched in metaphocytes logFC >2 (*Lin et al., 2019*); Orange: DGE of genes downregulated in metaphocytes logFC <2 (*Lin et al., 2019*). (E) Heat map showing the expression of metaphocyte signature genes. (F) Heat map showing the expression of phagocytosis and engulfment genes. (G) Venn diagram showing DGE between the three groups (logFC > |2|; FDR < 0.05).

The online version of this article includes the following figure supplement(s) for figure 5:

**Figure supplement 1.** Gating strategy for isolating *mpeg1*+ cells from juveniles.

**Figure supplement 2.** Expression profiles of non-hematopoietic and hematopoietic *mpeg1*+ cells of control and *csf1r^DM* juvenile zebrafish.

expressed at higher levels in zebrafish metaphocytes than in macrophages, LCs and neutrophils (*Lin et al., 2019*) (TPM logFC >2), and analyzed their expression in our data. This revealed that GFP⁺DsRed⁻ cells display a robust 'metaphocyte' gene signature (e.g. *cdh1*, *epcam*, *cldnh*, *cd4-1*), regardless of their genotype (*Figure 5D–E*). Additionally, many genes involved in phagocytosis and engulfment were downregulated in GFP⁺DsRed⁻ cells (e.g. *mertka*, *havcr1*, *stab1*, *Figure 5F*), as were genes that were previously shown to be expressed at lower levels in metaphocytes than in LCs and neutrophils (e.g. *itgb7*, *cdk1*, *cmklr1*, *cebpb*, *Figure 5—figure supplement 2B*). In line with the transcriptome similarities previously reported for metaphocytes and LCs, all cell populations in our analyses express *mpeg1* as well as genes related to antigen presentation (*mhc2dab*, *cd74a*, *cd83*) (*Figure 5—figure supplement 2A*). Together, these findings validate the qualification of skin GFP⁺DsRed⁻ cells as metaphocytes. Moreover, further analysis showed no major changes in the transcriptome of metaphocytes in the absence of *csf1r*, as only relatively few genes (359 out of 20.382) were found to differ significantly in expression between control and *csf1r^DM* GFP⁺DsRed⁻ cells (*Figure 5G*). Unexpectedly, many of these genes are involved in pigment cell differentiation. Taken together with our imaging analyses (*Figures 4* and *5A*), these data show that the skin of *csf1r^DM* zebrafish lack *mpeg1*+ macrophages, but exclusively contain *mpeg1*+ metaphocytes, which are not reliant on Csf1r-signaling.

## Csf1r^DM fish lack most mononuclear phagocytes

We wondered whether the macrophage deficiency observed in the skin represents a general feature of *csf1r^DM* fish. To address this question, we quantified total *mpeg1*+ cell numbers in 33 dpf and 1.5 months post fertilization (mpf) (juvenile zebrafish: between 30–90 dpf) control, *csf1r^DM* and *il34^-/-* fish by FACS (*Figure 5—figure supplement 1*). Fish deficient for *il34* were included as an extra control, since they exhibit a selective loss of branched skin macrophages and contain lower embryonic microglia numbers, but retain other macrophage populations (*Figure 4C*; *Kuil et al., 2019*; *Wu et al., 2018*). Indeed, *mpeg1*+ cell numbers, with macrophage scatter properties, obtained from whole *csf1r^DM* animals, were much lower than those in controls and *il34* mutants (*Figure 4—figure supplement 1A-B*). These findings are analogous to results reported for various organs of *Csf1r*-deficient mice and rats (*Dai et al., 2002*; *Pridans et al., 2018*). We next performed bulk RNA-sequencing on the total population of *mpeg1*+ cells isolated from controls, *csf1r^DM*, and *il34^-/-* (*Figure 6A*). PCA showed clustering of triplicates and segregation based on genotype (component 1: *csf1r^DM* versus controls/*il34^-/-*, component 2: *il34* mutants versus controls) (*Figure 6B*). In addition, gene expression profiling identified transcriptional programs consistent with phagocytic macrophages in control and *il34^-/-* *mpeg1*+ cells, but profiles consistent with metaphocytes only in *csf1r^DM* cells (*Figure 6C–E*). As overall *il34^-/-* animals have a relatively small and selective macrophage depletion, we argue that this could have prevented the detection of a metaphocyte signature. Collectively, this suggests that *csf1r^DM* fish specifically exhibit a profound deficiency in mononuclear phagocytes, whereas numerous remaining *mpeg1*+ cells appear to be metaphocytes rather than macrophages.

We further tested this possibility by lineage-tracing and surveyed, through flow cytometry, the presence of GFP⁺DsRed⁺ macrophages and GFP⁺DsRed⁻ metaphocytes among adult organs isolated from control and *csf1r^DM* fish. As previously reported, in the zebrafish brain, primitive hematopoiesis-derived *mpeg1*+ microglia are completely replaced by HSC-derived *mpeg1*+ cells, and therefore all adult microglia, as well as CNS-associated macrophages are GFP⁺DsRed⁺ (*Ferrero et al., 2018*). In addition, the lack of GFP⁺DsRed⁻ cells in the adult brain indicates that metaphocytes are not present in the central nervous system (*Figure 6F*). Brains of *csf1r^DM* zebrafish

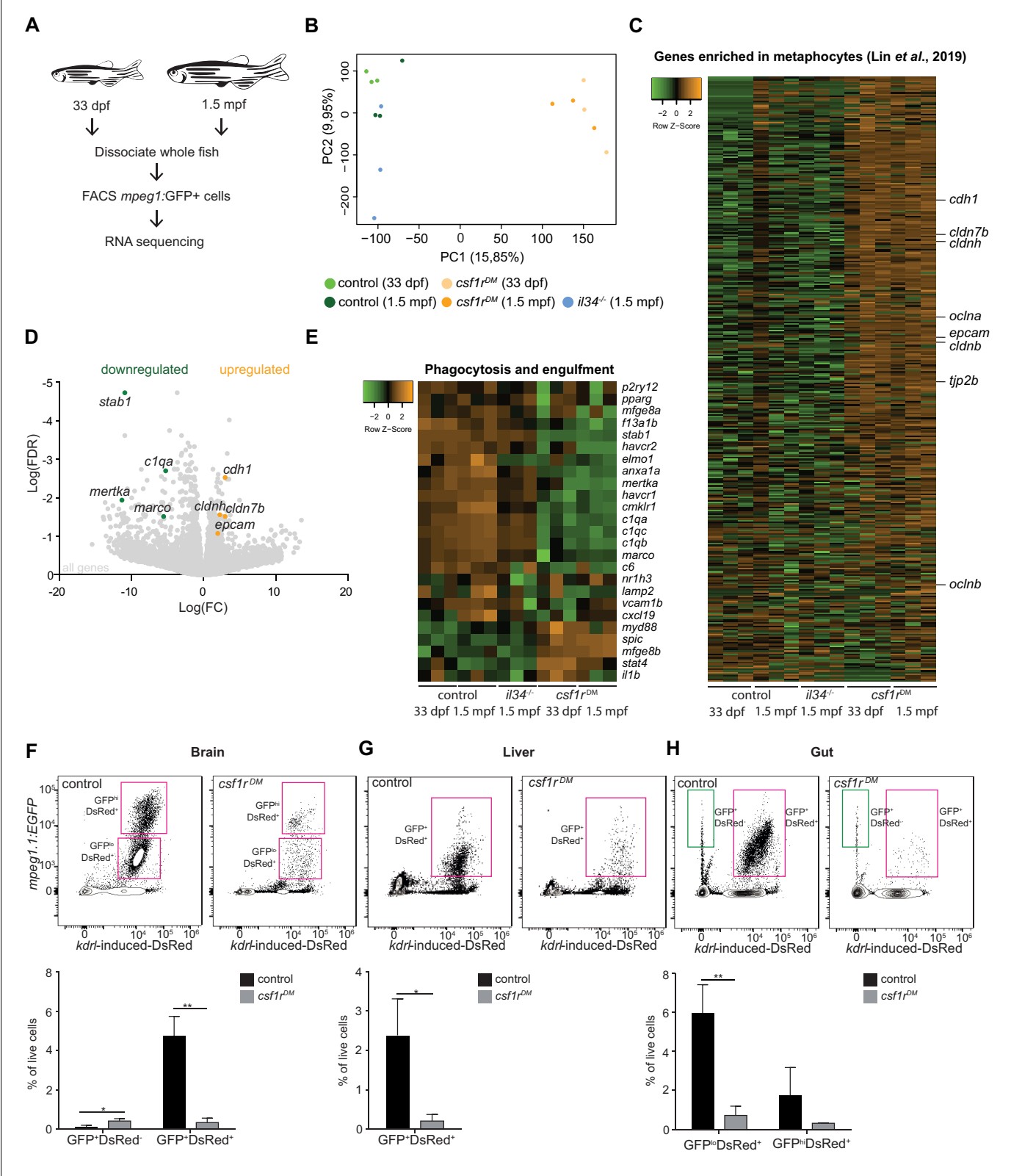

**Figure 6.** RNA sequencing of juvenile *mpeg1*+ cells and FACS analysis of brain, liver and gut, shows systemic depletion of macrophages in *csf1r*^DM zebrafish. (**A**) Schematic representation of the RNA sequencing strategy. (**B**) PCA analysis shows clustering of triplicates and segregation on genotype (control/*il34*^-/- vs. *csf1r*^DM). (**C**) Heat map showing the expression of metaphocyte signature genes in control, *il34*^-/- and *csf1r*^DM *mpeg1*+ cells. (**D**) Volcano plot showing gene expression changes between control and *csf1r*^DM at 1.5 mpf. Light grey: DGE of all geness, Green: DGE of some

*Figure 6 continued on next page*

*Figure 6 continued*

phagocytosis genes downregulated in *csf1r^DM mpeg1+* cells; Orange: DGE of genes enriched in metaphocytes (*Lin et al., 2019*). (E) Heat map showing phagocytosis and engulfment genes. (F–H) FACS analysis on cells from the adult (4 mpf) brain (F), liver (G) and gut (H) and quantifications. GFP⁺DsRed⁻=*mpeg1*+ only, GFP⁺DsRed⁺=*mpeg1*+/*kdrl*-induced-DsRed+.

were largely devoid of GFP⁺DsRed⁺ cells (*Figure 6F*), in line with our previous studies (*Oosterhof et al., 2019*; *Oosterhof et al., 2018*). Similarly, livers from control and *csf1r^DM* animals contained solely GFP⁺DsRed⁺ cells, which were virtually absent in *csf1r^DM* animals (*Figure 6G*). The intestine on the other hand contained both GFP⁺DsRed⁺ and GFP⁺DsRed⁻ cells (*Figure 6H*). However, these GFP⁺DsRed⁺ cells were lost and GFP⁺DsRed⁻ cell numbers were increased in *csf1r^DM*. As the presence of metaphocytes was reported in skin but also in the intestine (*Ferrero et al., 2020*; *Lin et al., 2019*), intestinal GFP⁺DsRed⁻ cells are likely also *csf1r*-independent metaphocytes. In all, *mpeg1+* macrophages are largely Csf1r-dependent, whereas *mpeg1+* cells present in the skin and intestine are Csf1r-independent non-hematopoietic metaphocytes (*Figure 7*).

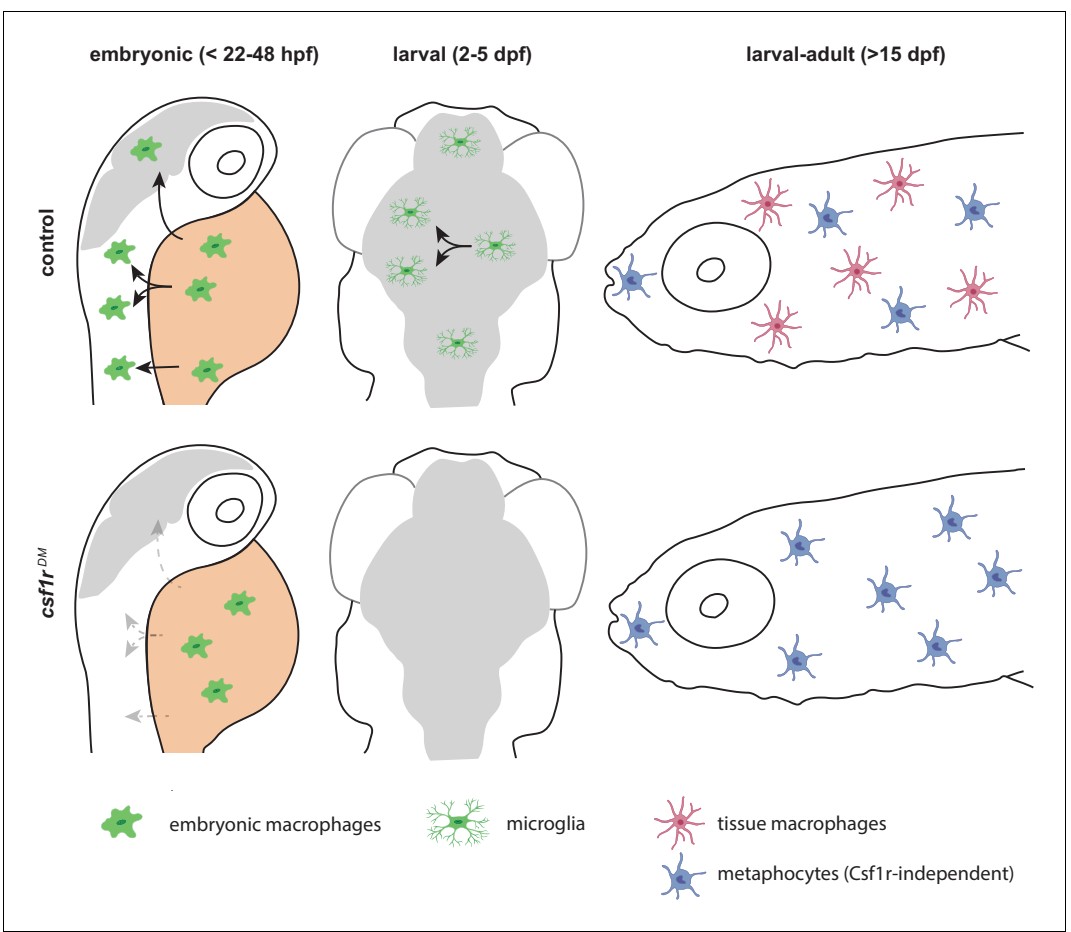

**Figure 7.** Schematic presentation of macrophage development in control and *csf1r*-deficient zebrafish. Upper panels indicate development of macrophages, microglia and definitive macrophages and metaphocytes: embryonic macrophages (left), microglia in larval brain (middle) and macrophages and metaphocytes in larva >15 dpf. Lower panels indicate abnormalities found in macrophage development in *csf1r*-deficient zebrafish: embryonic macrophages fail to migrate across the embryo (left), fewer macrophages arrive in the brain, and fail to divide (middle), metaphocytes develop normally whereas macrophages are depleted from larval to adult stages.

## Discussion

Here, we showed that embryonic macrophages, develop, proliferate, and also initially acquire macrophage behavior and gene expression profile independently of Csf1r. However, without functional Csf1r, these cells subsequently fail to distribute across the embryo and cease to expand in numbers. This phenotype explains particularly the strong effect on microglial precursors, as these invade the brain and expand in numbers early in embryonic development and microglia are absent throughout life in zebrafish, mice, rats and humans deficient for CSF1R. Around 15 days of age, however, a strong increase in *mpeg1+* macrophages in skin was detected by in vivo imaging in control but also in *csf1r*$^{DM}$ animals. Nevertheless, skin of both *csf1r*$^{DM}$ and mutants for the Csf1r ligand Il34 lacked the branched macrophages, which were present in controls, and only contained amoeboid *mpeg1+* cells. Based on their non-hematopoietic origin and shared transcriptome profile, we identified these cells as metaphocytes. As metaphocytes share markers, morphology, and gross behavior with macrophages, they are easily mistaken for macrophages. We further showed that *csf1r*$^{DM}$ adults lacked virtually all blood-derived *mpeg1+* mononuclear phagocytes, revealing the presence of *mpeg1+* metaphocytes in the gut, as well as in the skin. Our data shows that in zebrafish Csf1r is critical for generation of both embryonic and adult macrophages, but is dispensable for the development of metaphocytes. Therefore, *csf1r*-deficient zebrafish are macrophage-less in most organs, and as they are viable, enable us to study the in vivo consequences of the absence of macrophages for developmental and homeostatic cellular processes.

Two recent studies identified metaphocytes in zebrafish using distinct lineage tracing techniques, namely laser-mediated localized Cre-activation and CRISPR/Cas9 mediated genetic scarring followed by single cell DNA sequencing (*Alemany et al., 2018*; *Levraud and Herbomel, 2019*; *Lin et al., 2019*). Metaphocytes show reduced expression of engulfment genes, do not show a phagocytic response to injury or bacterial infection, have a rounded morphology and are highly motile (*Alemany et al., 2018*; *Lin et al., 2019*). Our transcriptome analysis showed high resemblance between metaphocytes and the remaining *mpeg1+* cells in *csf1r*$^{DM}$ zebrafish (total juvenile population and isolated from adult skin). Control and *csf1r*$^{DM}$ metaphocytes showed overall high similarity, but *csf1r*$^{DM}$ metaphocytes showed lower expression of genes involved in pigment cell differentiation. It is possible that this is an indirect consequence of the altered pigmentation status of *csf1r*$^{DM}$ deficient zebrafish, since they lack most of their xantophores, and lack stripes due to abnormal melanocyte patterning. As markers labeling macrophages will likely also label metaphocytes, this could perhaps explain the presumed incomplete depletion of macrophages in *Csf1r* mutant animals, or after CSF1R pharmacological inhibition (*Dai et al., 2002*; *Erblich et al., 2011*; *Pridans et al., 2018*). Even though, particularly in vitro, CSF1R is considered essential for macrophage development, macrophages are nevertheless detected, in numbers ranging between 10–70% of the numbers found in controls, in tissues, other than brain, epidermis and bone, of *Csf1r*-deficient mice and rats (*Dai et al., 2002*; *Pridans et al., 2018*). Therefore, at least in zebrafish, macrophage numbers in Csf1r-deficient mutants were initially overestimated (*Oosterhof et al., 2018*). As *CSF1R* mutations cause pleiotropic effects on various tissues in vertebrates and in human disease, that are likely caused by the absence of macrophages, our results further stress the importance of macrophages for development and homeostatic regulation of tissues. In addition, this raises the question whether metaphocytes exist in mammals (*Oosterhof et al., 2019*; *Oosterhof et al., 2018*).

In mouse *Csf1r* knockouts embryonic macrophages were reported to be largely absent from the yolk sac at E12.5 (*Ginhoux et al., 2010*). However, at E10.5 embryonic macrophages normally have already migrated away to the fetal liver and embryonic organs (*Stremmel et al., 2018*). Therefore, it is unknown whether primitive macrophages would be present in *Csf1r*-deficient mice at a stage earlier than E10.5 and can be generated independently of Csf1r. In *csf1r*$^{DM}$ fish we found initially normal embryonic macrophage numbers, but at 2–2.5 dpf, concordant with E12-13 in mice, we also found reduced macrophage numbers compared to controls. It remains to be determined whether CSF1R signaling is essential for embryonic development in mice and other mammals at earlier stages as well.

Homozygous mutations in *CSF1R* cause severe congenital brain disease with osteopetrosis, and absence of microglia (*Monies et al., 2017*; *Oosterhof et al., 2019*). Our data in zebrafish show multiple Csf1r-dependent steps of early microglia development that together illustrate how CSF1R-deficiency could underlie the absence of microglia already early in development. In zebrafish, only few

Csf1r-deficient microglial progenitors reach the developing brain, since they stop to expand, and they are unable to respond to neuronal expressed Interleukin-34, which normally facilitates brain colonization (*Greter et al., 2012*; *Kuil et al., 2018*; *Wang et al., 2012*; *Wu et al., 2018*). Thereafter, these few microglia do not expand, which eventually leads to their extinction. We propose that such a mechanism may underlie the absence of microglia, and osteoclasts, in patients with homozygous mutations in *CSF1R* (*Figure 7*).

We find in *il34*$^{-/-}$ zebrafish that branched skin macrophages are lacking, but we did not find substantially lower numbers of macrophages or obvious gene expression changes overall, as in *csf1r*$^{DM}$ zebrafish. This phenotype is reminiscent of that of *Il34* mutant mice that selectively lack microglia and Langerhans cells (*Greter et al., 2012*; *Wang et al., 2012*). Previous studies claimed skin *mpeg1* + hematopoietic branched cells in zebrafish to be Langerhans cells (*He et al., 2018*; *Lin et al., 2019*). It remains unclear whether these are true Langerhans cells, as there is no known zebrafish ortholog of langerin (CD207), the main marker of LCs in humans and mice. LCs are likely to exist in zebrafish, as Birbeck granules, the morphological markers of LCs, have been identified in zebrafish skin macrophages (*Lugo-Villarino et al., 2010*), and we recently demonstrated that zebrafish branched skin macrophages, develop independently of the transcription factor Irf8 (*Ferrero et al., 2020*), similar to mammalian LCs (*Chopin et al., 2013*; *Hambleton et al., 2011*). Their dependence on Il34 provides additional evidence for the conservation of LCs in zebrafish. The effect of Il34 loss on macrophage development is relatively subtle, and overall gene expression of *mpeg1*+ cells in *il34* mutants is likely to be dominated by gene expression from all Il34-independent macrophage populations and the effect of the loss of branched skin macrophages is therefore masked in the bulk RNA expression.

TRMs retain the ability to proliferate, partly due to the relief of transcriptional suppression of proliferative enhancers by MAFB (*Soucie et al., 2016*). Our findings suggest that Csf1r plays a central role in the maintenance of macrophage proliferative capacity. Our embryonic macrophage transcriptome analysis revealed two-fold lower expression of the majority of DNA replication genes in *csf1r*$^{DM}$ embryos, pointing towards a Csf1r-dependent induction of DNA replication, underlying the lack of macrophage proliferation. CSF1 can indeed rapidly stimulate S-phase entry and DNA replication of macrophages in vitro (*Tushinski and Stanley, 1985*). The Csf1r-independent proliferation of the earliest primitive macrophages on the yolk, could be explained by signaling through other members of the type III receptor tyrosine kinase family, including Csf3r, Flt3, or C-kit, of which two in zebrafish have been shown to be involved in the expansion of primitive macrophages (Flt3) or HSPCs (Kitb) (*Bartelmez and Stanley, 1985*; *He et al., 2014*; *Mahony et al., 2018*; *Sarrazin et al., 2009*; *Williams et al., 1992*). This could explain how the initial proliferation of progenitors is independent of Csf1r while later differentiation then becomes dependent.

In sum, our work provides new insight into the dynamics of embryonic and adult macrophage development, but also metaphocyte ontogeny in zebrafish, as well as the developmental requirements for Csf1r therein. The *csf1r*$^{DM}$ zebrafish are highly suitable for studying the effects of macrophage absence systemically and metaphocyte function in isolation. In addition, we provide an approach to discern Csf1r-independent metaphocytes from Csf1r-dependent macrophages. Our findings here provide insight into the mechanism that could also underlie the absence of microglia in *CSF1R*-related leukodystrophy and could help predict the effects on other TRM populations in response to CSF1R mutations or pharmacological inhibition.

## Materials and methods

**Key resources table**

| Reagent type (species) or resource | Designation | Source or reference | Identifiers | Additional information |
|---|---|---|---|---|
| Gene (*Danio rerio*) | Tg(mpeg1: EGFP)gl22 | *Ellett et al., 2011* | gl22Tg RRID:ZFIN_ZDB-ALT-120117-1 | Transgenic |

*Continued on next page*

*Continued*

| Reagent type (species) or resource | Designation | Source or reference | Identifiers | Additional information |
|---|---|---|---|---|
| Gene (*Danio rerio*) | *il34*^re03/re03 | **Kuil et al., 2019** | re03 RRID:ZFIN_ZDB-ALT-190814-11 | Mutant |
| Gene (*Danio rerio*) | *csf1rb*^re01/re01 | **Oosterhof et al., 2018** | re01 RRID:ZFIN_ZDB-ALT-180807-1 | Mutant |
| Gene (*Danio rerio*) | *csf1rb*^sa1503/sa1503 | ZIRC, This paper | sa1503 RRID:ZFIN_ZDB-ALT-120411-187 | Mutant |
| Gene (*Danio rerio*) | *csf1ra*^j4e1/j4e1 | **Parichy et al., 2009** | j4e1 RRID:ZFIN_ZDB-ALT-001205-14 | Mutant |
| Gene (*Danio rerio*) | *Et(shhb:KalTA4,UAS-E1b:mCherry)*^zf279 | (**Distel et al., 2009**) | zf279Et RRID:ZFIN_ZDB-ALT-120221-7 | Transgenic |
| Gene (*Danio rerio*) | Tg(kdrl:Cre)^s898 | **Bertrand et al., 2010** | s898Tg RRID:ZFIN_ZDB-ALT-100419-3 | Transgenic |
| Gene (*Danio rerio*) | *Tg(actb2:loxP-STOP-loxP-DsRed*^express)sd5 | **Bertrand et al., 2010** | sd5Tg RRID:ZFIN_ZDB-ALT-100301-1 | Transgenic |
| Antibody | anti-PCNA (mouse monoclonal) | Agilent | Agilent Cat# M0879, RRID:AB_2160651 | IHC (1:250) |
| Antibody | Anti-DsRed (rabbit polyclonal) | Takara Bio Clontech | Takara Bio Cat# 632496, RRID:AB_10013483 | IHC (1:500) |
| Antibody | Anti-GFP (chicken polyclonal) | Abcam | Abcam Cat# ab13970, RRID:AB_300798 | IHC (1:500) |
| Antibody | anti-Lplastin (rabbit) | gift from Yi Feng, University of Edinburgh | | IHC (1:500) |
| Commercial assay or kit | Click-iT EdU | Invitrogen | C10340 | Cell Proliferation Kit for Imaging |
| Software, algorithm | Prism 5 | Graphpad | GraphPad Prism, RRID:SCR_002798 | Data visualization and statistics software |
| Software, algorithm | Leica | LASX | Leica Application Suite X, RRID:SCR_013673 | Microscope image processing software |
| Software, algorithm | FIJI | ImageJ | National Center for Microscopy and Imaging Research: ImageJ Mosaic Plug-ins, RRID:SCR_001935 | Image analysis software |
| Software, algorithm | FlowJo v10 | Treestar | FlowJo, RRID:SCR_008520 | FACS software |
| Software, algorithm | R (Bioconductor package) | **Durinck et al., 2009**; **Robinson et al., 2010** | edgeR, RRID:SCR_012802 GAGE, RRID:SCR_017067 | Transcriptomics data analysis software |

## Animals

Zebrafish deficient for both Csf1ra (*csf1ra*^j4e1/j4e1) and Csf1rb (*csf1rb*^re01/ re01), *csf1r*^DM, were used as we described previously (**Oosterhof et al., 2018**). The *csf1ra*^j4e1/j4e1 mutant was combined with a second *csf1rb* allele, *csf1rb*^sa1503/sa1503, affecting an essential splice site, leading to a premature

STOP codon, for flow cytometry and lineage tracing experiments. Zebrafish deficient in Csf1a/Csf1b (*csf1a*^re05/re05^; *csf1b*^re07/re07^) or Il34 (*il34*^re03/re03^) are described previously (**Kuil et al., 2019**). Tg (*mpeg1:egfp*); *Et(shhb:KalTA4,UAS-E1b:mCherry)*^zf279^) were used as control animals (**Ellett and Lieschke, 2010**; **van Ham et al., 2014**). *For the genetic lineage tracing the following transgenic lines were crossed:* Tg(*kdrl*:Cre)^s898^ and Tg(*actb2:loxP-STOP-loxP-DsRed*^express^)^sd5^ (**Bertrand et al., 2010**). All control animals used throughout the manuscript are *wild-type* controls carrying the trangene reporter constructs only. Adult and larval fish were kept on a 14h/10h light–dark cycle at 28°C. Larvae were kept in HEPES-buffered E3 medium. Media was refreshed daily and at 24 hpf 0.003% 1-phenyl 2-thiourea (PTU) was added to prevent pigmentation. Animal experiments were approved by the Animal Experimentation Committees of the Erasmus MC and ULB.

## Live imaging

Intravital imaging in zebrafish brains was largely performed as previously described (**van Ham et al., 2014**). Briefly, zebrafish larvae were mounted in 1.8% low melting point agarose containing 0.016% MS-222 as sedative and anesthetic in HEPES-buffered E3. The imaging dish containing the embedded larva was filled with HEPES-buffered E3 containing 0.016% MS-222.

For the experiment where larvae were followed over time between 5 and 9 dpf, larvae were removed from the low melting point agarose after imaging and put individually in wells of a 6 wells-plate containing HEPES-buffered E3 with PTU in which they were fed paramecia.

For the experiment with larval fish between 8 and 24 dpf fish were kept in E3 medium until 5 dpf. From 5 dpf onwards, *wild-type* controls, *il34,* and *csf1r* mutants were raised under standard conditions (14h/10h light–dark cycle, 28°C) in the aquaria (Tecniplast, Italy) in the Erasmus MC fish facility and fed paramecia and dry food. From 13 dpf onwards they were also fed brine shrimp. Animals from all experimental groups were raised with the same number of fish per tank, in tanks of the same size throughout the experiment. Confocal imaging was performed using a Leica SP5 intravital imaging setup with a 20x/1.0 NA water-dipping lens. Imaging of *mpeg1*-GFP was performed using the 488 nm laser. Analysis of imaging data was performed using imageJ (FIJI) and LAS AF software (Leica).

## Immunofluorescence staining

Immunohistochemistry was performed as described (**van Ham et al., 2014**; **van Ham et al., 2012**). Briefly, larvae were fixed in 4 % PFA at 4°C overnight. Subsequently, they were dehydrated with an increasing methanol concentration methanol series, stored in 100% methanol at -20°C for at least 12 hours, and rehydrated, followed by incubation in 150 mM Tris-HCl (pH=9.0) for 15 minutes at 70°C. Samples were then washed in PBS containing 0.04% Triton (PBST) and incubated in acetone for 20 minutes at -20°C. After washing in PBST and ddH₂O, larvae were incubated for three hours in blocking buffer (10 % goat serum, 1 % Triton X-100 (Tx100), 1% BSA, 0.1 % Tween-20 in PBS) at 4°C, followed by incubation in primary antibody buffer at 4°C for three days. Larvae were washed in 10 % goat serum 1 % Tx100 in PBS and PBS containing 1 % TX100 for a few hours, followed by incubation in secondary antibody buffer at 4°C for 2.5 days. Hereafter the secondary antibody was washed away using PBS. Primary antibody buffer: 1 % goat serum, 0.8 % Tx100, 1 % BSA, 0.1 % Tween-20 in PBS. Secondary antibody buffer: 0.8 % goat serum, 1 % BSA and PBS containing Hoechst. Primary antibodies: PCNA (1:250, Dako), L-plastin (1:500, gift from Yi Feng, University of Edinburgh). Secondary antibodies used were DyLight Alexa 488 (1:250) and DyLight Alexa 647 (1:250). Samples were imaged as described above.

## Immunostaining of fish scales

Scales were manually detached from anesthetized fish and pre-treated with 100mM DTT (Invitrogen) before O/N fixation in 4 % PFA. Immunostaining on floating scales was performed as described, using the following primary and secondary antibodies: chicken anti-GFP polyclonal antibody (1:500; Abcam), rabbit anti-DsRed polyclonal antibody (1:500; Clontech), Alexa Fluor 488-conjugated anti-chicken IgG antibody (1:500; Invitrogen), Alexa Fluor 594-conjugated anti-rabbit IgG (1:500; Abcam). Images were taken with a Zeiss LSM 780 inverted microscope, using a Plan Apochromat 20× objective. Image post-processing (contrast and gamma adjust) were performed with the Zeiss Zen Software.

## EdU pulse-chase protocol

Larvae of 4 dpf were placed in a 24 wells plate in HEPES buffered (pH = 7.3) E3 containing 0.003% PTU and 0.5 mM EdU for 24 hours. Next, larvae were fixed in 4% PFA for 3 hours at room temperature, dehydrated with a 25%, 50%, 75%, 100% MeOH series and stored at -20°C for at least 12 hours. Rehydrated in series followed by a proteinase K (10 μg/ml in PBS) incubation for an hour. Followed by 15 minute post fixation in 4% PFA. Larvae were further permeabilized in 1% DMSO in PBS-T. Thereafter 50μl Click-iT (Invitrogen) reaction cocktail was added for 3 hours at room temperature protected from light. After washing steps larvae were subjected to immunolabelling using L-plastin (see section immunofluorescent labelling). Samples were imaged as described above.

## Quantification of live-imaging data and stainings

The number of cells was manually quantified using ImageJ (FIJI) or Leica LASX software. To generate an overview of the gross migratory patterns maximum intensity projections of timelapse recordings were generated in FIJI.

## Isolation of *mpeg1*-GFP+ cells from zebrafish larvae and adult fish

At 28 hpf, 35 larvae were collected in 0.16 % MS-222 solution to euthanize them before adding 5x Trypsin-EDTA (0.25% Trypsin, 0.1 % EDTA in PBS). For *csf1r*$^{DM}$ cells, at 50 hpf, 70 larvae were used as these mutants had fewer *mpeg1*-GFP positive cells. Micro centrifuge tubes containing zebrafish embryos were incubated on ice on a shaking platform to dissociate the cells. At 33 dpf and 1.5 mpf, single fish were euthanized in ice water, imaged to measure their length, and they were cut in small pieces with a razor blade and incubated in 5x Trypsin-EDTA on ice for 1 hour to dissociate. Next, the cell suspension was transferred to FACS tubes by running it over a 35 μm cell strainer cap. PBS containing 10 % fetal calf serum (FCS) was added over the strainer caps and the samples were centrifuged for 10 minutes 1000 rpm at 4°C. The pellet was taken up in 300 μl PBS-10% FCS containing DAPI (1:1000). After analysis based on myeloid scatter, singlets, dapi and *mpeg1*-GFP signal cells were FAC-sorted by FACSAria IIIu and collected in Trizol, followed by RNA isolation according to the manufacturer's instructions (SMART-Seq v4 Ultra Low Input RNA Kit for Sequencing, Takara Bio USA) (*Figure 5—figure supplement 1*). Single-cell suspensions of dissected adult zebrafish organs were prepared as previously described (*Wittamer et al., 2011*). Flow cytometry and cell sorting were performed with a FACS ARIA II (Becton Dickinson). For RNA-sequencing, *mpeg1*-GFP-positive cells from the skin were collected in Qiazol and RNA was extracted using the miRNeasy Micro Kit (Qiagen). Analyses were performed using the FlowJo software (Treestar).

RNA sequencing cDNA was synthesized and amplified using SMART-seq V4 Ultra Low Input RNA kit for Sequencing (Takara Bio USA, Inc) following the manufacturer's protocol. Amplified cDNA was further processed according to TruSeq Sample Preparation v.2 Guide (Illumina) and paired end-sequenced (2×75 bp) on the HiSeq 2500 (Illumina). Experiment 1, embryonic macrophages were sequenced at between 12 and 21 million reads per sample. Experiment 2, juvenile macrophages, were sequenced at between 5 and 106 million reads per sample. Reads were mapped using Star v2.5 against the GRCz10 zebrafish genome (*Dobin et al., 2013*). For differential gene expression analysis and GSEA we used the Bioconductor packages edgeR and GAGE, respectively (*Durinck et al., 2009*; *Luo et al., 2009*; *Robinson et al., 2010*).

For analyses on adult skin *mpeg1+* cells, RNA quality was checked using a Bioanalyzer 2100 (Agilent technologies). Indexed cDNA libraries were obtained using the Ovation Solo RNA-Seq System (NuGen-TECAN) with the SoLo Custom AnyDeplete Probe Mix (Zebrafish probe set) following

**Table 1.** List of primers used for qPCR experiments.

| Gene | Forward Primer | Reverse Primer |
|---|---|---|
| *ef1α* | GAGAAGTTCGAGAAGGAAGC | CGTAGTATTTGCTGGTCTCG |
| *mob4* | CACCCGTTTCGTGATGAAGTACAA | GTTAAGCAGGATTTACAATGGAG |
| *csf1a* | ACGTCTGTGGACTGGAACTG | CTGTTGGACAAATGCAGGGG |
| *csf1b* | GGATTTGGGTCGGTGAGCTT | TGGAGAGGGGAACACACAGT |
| *il34* | AGGGAGTTTCCGACGCTTTT | CTGAGAAGCCAGCATTCGGA |

**Table 2.** Number of biological replicates per group for qPCR.

| Age | csf1a | csf1b | il34 |
|---|---|---|---|
| 20 hpf | 3 | 1 | |
| 24 hpf | 5 | 2 | 1 |
| 36 hpf | 5 | 5 | 4 |
| 48 hpf | 3 | 4 | 2 |
| 72 hpf | 4 | 4 | 3 |
| 7 dpf | 5 | 4 | 5 |
| 10 dpf | 4 | 4 | 4 |
| 14 dpf | 3 | 3 | 2 |
| **Organ** | | | |
| Gills | 3 | 3 | 3 |
| Skin | 3 | 4 | 3 |
| Muscle | 4 | 3 | 2 |
| Kidney | 4 | 4 | 2 |
| Heart | 5 | 2 | 4 |
| Spleen | 3 | 2 | 2 |
| Eye | 5 | 5 | 3 |
| Brain | 6 | 6 | 5 |
| Liver | 4 | 2 | 5 |
| Intestine | 3 | 1 | 3 |

manufacturer recommendation. The multiplexed libraries were loaded on a NovaSeq 6000 (Illumina) using a S2 flow cell and sequences were produced using a 200 Cycle Kit. On average 65 million paired-end reads were mapped against the *Danio rerio* reference genome GRCz11.94 using STAR software to generate read alignments for each sample. Annotations Danio_rerio.GRCz11.94.gtf were obtained from ftp.Ensembl.org. After transcripts assembling, gene level counts were obtained using HTSeq. Genes differentially expressed were identified used the Bioconductor packages edgeR (*Durinck et al., 2009*; *Robinson et al., 2010*).

## qPCR

Relative amount of each transcript was quantified via the *ΔCt method,* using *MOB family member 4 (mob4)* or *elongation-Factor-1-alpha (ef1α)* expression for normalization, or via the ΔΔCt method, using *mob4* or *ef1α* and WKM for normalization. Primers are listed in *Table 1*. The number of biological replicates are listed in *Table 2*.

## Statistical analysis

For statistical analysis GraphPad was used to perform Student's *t*-tests, one-way ANOVA with Dunnett's multiple comparison test, linear regression and non-linear regression analysis. Results were regarded significant at $p < 0.05$.

## Acknowledgements

We acknowledge Remco Hoogenboezem, Frederick Libert and Anne Lefort for assistance in RNA sequencing analysis and Michael Vermeulen for assistance with FACS. We thank Leslie Sanderson and Stefan Barakat for helpful comments on the manuscript, and the optical imaging center (OIC) of the Erasmus MC for assistance in confocal microscopy. Research in the van Ham laboratory is supported by a Marie Curie Career Integration Grant, a ZonMW VENI grant and an Erasmus University Rotterdam fellowship. Work in the Wittamer lab is funded by the WELBIO, the Fonds de la Recherche Scientifique FNRS under Incentive Grant for Scientific Research and The Minerve

Foundation. GF is a FNRS Research Fellow. The work in Petr Bartunek lab was funded by the Czech Science Foundation, grant number 18–18363S.

## Additional information

### Funding

| Funder | Grant reference number | Author |
|---|---|---|
| Erasmus University Rotterdam | University Fellowship | Tjakko J van Ham |
| WELBIO | WELBIO-CR-2015S-04 | Valerie Wittamer |
| Marie Curie Career Integration Grant | 322368 | Tjakko J van Ham |
| ZonMw | VENI 016.136.150 | Tjakko J van Ham |
| Fonds De La Recherche Scientifique - FNRS | F451218F | Valerie Wittamer |
| Czech Science Foundation | 18-18363S | Tereza Mikulášová |

The funders had no role in study design, data collection and interpretation, or the decision to submit the work for publication.

### Author contributions

Laura E Kuil, Conceptualization, Formal analysis, Supervision, Investigation, Visualization, Methodology, Writing - original draft, Project administration, Writing - review and editing; Nynke Oosterhof, Software, Investigation, Methodology, Writing - original draft; Giuliano Ferrero, Tereza Mikulášová, Martina Hason, Mireia Rovira, Erik MJ Bindels, Resources, Investigation, Methodology; Jordy Dekker, Formal analysis, Investigation, Methodology; Herma C van der Linde, Conceptualization, Resources, Formal analysis, Investigation, Methodology; Paulina MH van Strien, Emma de Pater, Resources, Methodology; Gerben Schaaf, Methodology, Writing - review and editing; Valerie Wittamer, Conceptualization, Resources, Formal analysis, Supervision, Funding acquisition, Investigation, Methodology, Writing - review and editing; Tjakko J van Ham, Conceptualization, Supervision, Funding acquisition, Investigation, Visualization, Writing - original draft, Writing - review and editing

### Author ORCIDs

Laura E Kuil (iD) https://orcid.org/0000-0003-3330-1371
Gerben Schaaf (iD) http://orcid.org/0000-0003-0189-9073
Valerie Wittamer (iD) https://orcid.org/0000-0003-0003-2646
Tjakko J van Ham (iD) https://orcid.org/0000-0002-2175-8713

### Ethics

Animal experimentation: All experimental procedures were approved and in accordance with the recommendations of the Animal Experimentation Committee at Erasmus MC, Rotterdam or the UL Bethical committee for animal welfare (CEBEA) (protocol 594N). Zebrafish over 5 days old were euthanized using ice water and/or high dose MS-222.

### Decision letter and Author response

Decision letter https://doi.org/10.7554/eLife.53403.sa1
Author response https://doi.org/10.7554/eLife.53403.sa2

## Additional files

### Supplementary files

- Transparent reporting form

## Data availability

The data discussed in this publication have been deposited in NCBI's Gene Expression Omnibus (Edgar et al., 2002) and are accessible through GEO Series accession number GSE149789.

The following dataset was generated:

| Author(s) | Year | Dataset title | Dataset URL | Database and Identifier |
|---|---|---|---|---|
| Kuil LE, Oosterhof N, Ferrero G, Mikulášová T, Hason M, Dekker J, Rovira M, van der Linde HC, van Strien PMH, Pater E, Schaaf G, Bindels EMJ, Wittamer V, Ham TJ | 2020 | Zebrafish mpeg+ cells | https://www.ncbi.nlm.nih.gov/geo/query/acc.cgi?acc=GSE149789 | NCBI Gene Expression Omnibus, GSE149789 |

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
