## [Decision Letter]

**Acceptance summary:**

Kuil et al. outline the phenotypic consequences on macrophages/myeloid cells following csf1r depletion in zebrafish, demonstrating the essential roles played by *csf1r* in macrophage development and proliferation. On the other hand, *csf1r* is not required for the development of myeloid-like metaphocytes, recently discovered ectoderm-derived cells in zebrafish.

**Decision letter after peer review:**

Thank you for submitting your article "Macrophage developmental arrest in zebrafish reveals the emergence of Csf1r-1 independent myeloid-like cells" for consideration by *eLife*. Your article has been reviewed by Didier Stainier as the Senior Editor, a guest Reviewing Editor, and three reviewers. The reviewers have opted to remain anonymous.

The reviewers have discussed the reviews with one another and the Reviewing Editor has drafted this decision to help you prepare a revised submission.

Summary:

In the submitted manuscript, Kuil et al., outline the phenotypic consequences on macrophages/myeloid cells following *csf1r* depletion in zebrafish. The authors find that csf1r loss results in several issues in macrophage development and proliferation, culminating in supposedly ectoderm-derived MLCs. These findings mirror similar observations in other models and add further observations to a possible ectodermal origin of myeloid-like cells in zebrafish.

The manuscript is well-written and provides an interesting characterization that could be the basis of further investigations on the molecular mechanisms underlying the emergence of MLCs in a *csf1r*-independent manner.

Overall, the manuscript contains a wealth of data, which in particular in the first half of the manuscript is well presented. However, the manuscript's precision deteriorates somewhat in the latter parts, when the authors lean back on inferences and at times vague phenotype descriptions.

Essential revisions:

1) Throughout the paper, the authors leave one major question unanswered – where are the Csf1R ligands, i.e. Il34, Csf1a and Csf1b, expressed? This becomes especially pressing after the authors summarize the dynamics observed in wt vs csf1rDM (subsection “*sf1r*DM skin lacks highly branched Langerhans cells”). Adding this data would possibly provide further insight into the observed phenotypes in csf1rDM animals.

2) Proliferation/cell division defects are assessed by quantification in the first paragraph of the result section. The authors perform EdU labelling experiments between 4 to 5 dpf and focus their analysis only on microglia. The authors should analyze proliferation not just by manual quantification but with a proper experiment (as EdU labelling) between 32-48 hpf and 56-72 hpf as well). Moreover, in this analysis the authors should not only focus on microglia, but consider the entire embryo.

3) Subsection “*sf1r*DM skin lacks highly branched Langerhans cells”, the authors speculate that branched macrophages absent in csf1rDM animals may represent Langerhans cells. The authors maintain this speculation but turn it increasingly into matter-of-fact language with progression of their manuscript. Similarly, the authors already squarely refer to the cells as de facto Langerhans cells in the paper abstract. This inference should be dampened considerably. In fact, the authors conclude they identified Langerhans cells based on location, morphology and behavior. It is likely, but are there other ways for the authors to support this conclusion? Would it be possible for the authors to check the expression of markers, by ISH, by IF, by reporter lines, to confirm the identity of these cells?

4) At times, the authors are pretty liberal in their claims what individual citations show, and at times cherry-pick data points seemingly out of context. These should be revisited. Case in point, the statement that genetic lineage tracing established a non-hematopoietic origin of csf1rDM mpeg1+ cells (Introduction) is attributed to Alemany et al., 2018 and Ferrero et al., 2018 – these papers however establish a) that by single-cell seq, some myeloid cells seem different than canonical ones, (first ref) and b) focused on microglia specifically, that turns out to be all coming from mesodermal/blood origins (second ref).

5) The identification of the csf1r-independent mpeg1+ cells as MLCs is too much based on speculations. For example, showing that such cells are not lineage traced with Kdrl-Cre is far away from demonstrating that the cells are of ectodermal origin, as MLCs are supposed to be. If the main message of the paper is that MLCs are csf1r-independent, then the authors need to positively label MLCs with an ectoderm-specific lineage tracing tool or reporter line. Otherwise SP cells in the adult mutants could be pMacs that expanded with delay in the mutants, or even macrophages derived from EMPs or HSCs that escaped Cre/Lox recombination, which is never 100% efficient. Can the authors provide any better evidence for their suggested conclusions?

6) On this front, at row subsection “Remaining *mpeg1+* cells in *csf1r*DM skin are macrophage-like cells (MLCs) “the authors say that 9% of these canonical macrophage genes were downregulated in csf1rDM mpeg1+ cells. I find that 9% is a surprisingly small difference in gene expression for being cells of different origin/lineage.

7) "MLCs are ectoderm-derived, do not express csf1ra and csf1rb, are mpeg1+, express many genes also expressed in Langerhans cells/macrophages, but lack a phagocytic response upon infection or injury (Alemany et al., 2018; Lin et al., 2019)". Subsection “*csf1r*DM skin lacks highly branched Langerhans cells”. Can the authors test that the MLCs present in their mutants are actually affected in their phagocytic response? Could they perform injury assay in juvenile fish?

8) The authors conclude that "mpeg1+ cells that remain in csf1rDM zebrafish skin are, based on gene expression, morphology, cell migration and lineage tracing very likely MLCs.". The gene expression analysis is based on qPCR analysis of few selected genes on SP and DP cells. The existing molecular characterisation of the putative csf1r-independent MLCs needs being implemented by at least validating cdh1 and cldn7b, as examples of MLC genes, via qPCR and by showing the pattern of cldnh1 in the RNAseq. The pattern of such MLC markers should also be assessed in SP pMacs at earlier stages as a negative control. If possible, given the expertise shown in this paper, the authors should perform RNA-seq on FACS sorted SP (control and mutant) and DP (control) cells. MLCs are recently described cells and we still know very little about them. I think this paper, the group and others in the community would benefit in the generation of such a data set.

---

## [Author Response]

Essential revisions:1) Throughout the paper, the authors leave one major question unanswered – where are the Csf1R ligands, i.e. Il34, Csf1a and Csf1b, expressed? This becomes especially pressing after the authors summarize the dynamics observed in wt vs csf1rDM (subsection “sf1rDM skin lacks highly branched Langerhans cells”). Adding this data would possibly provide further insight into the observed phenotypes in csf1rDM animals.

The three ligands are expressed at very low levels, and we and others have not been able to consistently detect expression in larvae by mRNA in situ hybridization. In juvenile/adult zebrafish, expression of both csf1a and csf1b has been detected in skin, more specifically in interstripe iridophores and hypodermal and fin cells (Patterson et al., 2013). We have added this information to the text. We have also performed additional expression analyses and included expression data of all three ligands in embryonic stages and several adult organs (Figure 2—figure supplement 1C-D). We have also added the following sections to the text:

Subsection “Impaired proliferation of embryonic *csf1rDM* 177 macrophages is reflected in their transcriptome and proliferation is not restored in microglia”: Of the three Csf1r ligand genes, both csf1a and csf1b are expressed at 20 hpf, whereas il34 is not detectable at that time, barely detectable at 24 hpf, and moderately expressed at 36 hpf (Figure S2C). Therefore, it is possible that the reduced expression of cell cycle related genes in csf1r^DM^ macrophages could be attributed largely to a lack of interaction between the two Csf1 ligands and Csf1r. Additionally, this suggests that these two ligands likely do not influence the specification of embryonic macrophages at this stage. Previous analyses of macrophage development in il34^-/-^ deficient zebrafish around 30 hpf showed primarily a deficiency in the migration of macrophages across the embryo and into the brain (Kuil et al., 2019; Wu et al., 2018).

Subsection “*csf1r*DM 220 skin lacks highly branched putative Langerhans cells”: In larval zebrafish, csf1a and csf1b expression were detected in skin (Figure 2—figure supplement 1D), more specifically in interstripe iridophores and hypodermal and fin cells (Patterson and Parichy, 2013). Although we found that il34 was also expressed in adult skin, this expression was about 10-fold lower than that of csf1a or csf1b (Figure 2—figure supplement 1D). However, our in vivo imaging data suggests that the loss of Il34, but not of both Csf1a and Csf1b, affects branched skin macrophages in particular (Figure 4—figure supplement 1B).

2) Proliferation/cell division defects are assessed by quantification in the first paragraph of the result section. The authors perform EdU labelling experiments between 4 to 5 dpf and focus their analysis only on microglia. The authors should analyze proliferation not just by manual quantification but with a proper experiment (as EdU labelling) between 32-48 hpf and 56-72 hpf as well). Moreover, in this analysis the authors should not only focus on microglia, but consider the entire embryo.

We have RNA sequenced embryonic macrophages at both 1 and 2 dpf, containing only few early microglia. We find at both time points strongly decreased expression of various classes of proliferation-related genes, including cell-cycle regulators and components of the DNA replication machinery. This is consistent with the early studies on CSF1R function, where addition of CSF1 to macrophages in culture showed massively increased DNA and proliferation (Tushinski and Stanley, 1983, 1985). Regarding peripheral macrophages, we literally see cells dividing in controls -but not in the csf1r-mutants- by in vivo imaging, explaining the increase in macrophage numbers. As we could not directly see this for microglia, and to address this in detail for a specific population of macrophages, we performed EdU staining of early microglia. We have attempted previously to perform combined Lplastin/EdU staining on the yolk ball and had difficulty getting this to work. Additionally, with the above in mind, we do not see how such an experiment would add substantially to our conclusions or what this would control for.

3) Subsection “sf1rDM skin lacks highly branched Langerhans cells”, the authors speculate that branched macrophages absent in csf1rDM animals may represent Langerhans cells. The authors maintain this speculation but turn it increasingly into matter-of-fact language with progression of their manuscript. Similarly, the authors already squarely refer to the cells as de facto Langerhans cells in the paper abstract. This inference should be dampened considerably. In fact, the authors conclude they identified Langerhans cells based on location, morphology and behavior. It is likely, but are there other ways for the authors to support this conclusion? Would it be possible for the authors to check the expression of markers, by ISH, by IF, by reporter lines, to confirm the identity of these cells?

We agree with the reviewer that the location, morphology and behavior of branched mpeg1 cells do not constitute definitive proof that these are Langerhans cells. We were influenced in this regard by a previous manuscript in *eLife*, where the authors based on similarly limited criteria coined these cells Langerhans cells (He et al., 2018). Whether these are true Langerhans cells is actually difficult to address, as there is no known zebrafish ortholog of langerin (CD207), the protein that defines Langerhans cells in humans and mice. There is no doubt about the existence of this cell population in zebrafish, however, as Birbeck granules, the morphological markers of mammalian Langerhans cells, have been previously identified in zebrafish skin macrophages by transmission electron microscopy (Lugo et al., 2010). Moreover, the branched cells we describe in this study share several other properties with Langerhans cells, including dependence on interleukin-34, as shown here, and independence on the transcription factor irf8, as we previously reported (Ferrero et al., 2020). Nevertheless, it is not a main point of our paper what type of subpopulation of macrophages this is. As suggested, we have thus toned down our conclusion and adjusted the text to clarify that this is a speculation.

4) At times, the authors are pretty liberal in their claims what individual citations show, and at times cherry-pick data points seemingly out of context. These should be revisited. Case in point, the statement that genetic lineage tracing established a non-hematopoietic origin of csf1rDM mpeg1+ cells (Introduction) is attributed to Alemany et al., 2018 and Ferrero et al., 2018 – these papers however establish a) that by single-cell seq, some myeloid cells seem different than canonical ones, (first ref) and b) focused on microglia specifically, that turns out to be all coming from mesodermal/blood origins (second ref).

We thank the reviewer for catching these errors, which we have now corrected. We have also made sure that all references are valid and properly cited.

5) The identification of the csf1r-independent mpeg1+ cells as MLCs is too much based on speculations. For example, showing that such cells are not lineage traced with Kdrl-Cre is far away from demonstrating that the cells are of ectodermal origin, as MLCs are supposed to be. If the main message of the paper is that MLCs are csf1r-independent, then the authors need to positively label MLCs with an ectoderm-specific lineage tracing tool or reporter line. Otherwise SP cells in the adult mutants could be pMacs that expanded with delay in the mutants, or even macrophages derived from EMPs or HSCs that escaped Cre/Lox recombination, which is never 100% efficient. Can the authors provide any better evidence for their suggested conclusions?

We agree that our system is not perfect and that a precise characterization of csf1r-independent mpeg1+ cells would strengthen our point. For this purpose, we performed additional RNA sequencing experiments (New Figure 5) on the different populations of interest (skin GFP+/DsRed- and GFP+/DsRed+ cells isolated from control and csf1r^DM^ fish). We found that these populations exhibit distinct signatures consistent with being macrophages (GFP+/DsRed+) or metaphocytes (GFP+/DsRed-). Importantly, the transcriptome of GFP+/DsRed- cells is almost indistinguishable from that of metaphocytes shown by Lin et al., 2019. This comparison can be made since they also compared the metaphocyte gene expression with that of “Langerhans cells“ (that is, other macrophages in the scales). Therefore, we believe that these new experiments definitively validate the metaphocyte identity of mpeg1+ cells in csf1r mutants and GFP+/DsRed- cells in our model. We have now added these data in the revised version (See changes in the manuscript and Figure 5 and Figure 6 and Figure 5—figure supplement 2).

In our recent report we show that the frequency of skin GFP+/DsRed- cells also remains unchanged in irf8 knockout fish, which completely lack primitive macrophages (Ferrero et al., 2020). Although we cannot exclude a minor contribution of primitive hematopoiesis, this strongly suggests that metaphocytes make up for the majority of the GFP+DsRed- population present in the skin of adult Tg(kdrl:cre; bactin:switch; mpeg1:GFP) transgenics.

6) On this front, at row subsection “Remaining mpeg1+ cells in csf1rDM skin are macrophage-like cells (MLCs) “the authors say that 9% of these canonical macrophage genes were downregulated in csf1rDM mpeg1+ cells. I find that 9% is a surprisingly small difference in gene expression for being cells of different origin/lineage.

We agree that this is a small number for being cells of different origin. That said, the remarkable feature of metaphocytes according to Li et al., 2019 is that despite this difference in lineage, their gene expression overlaps largely with macrophages. Our new RNAseq data on are consistent with this conclusion.

7) "MLCs are ectoderm-derived, do not express csf1ra and csf1rb, are mpeg1+, express many genes also expressed in Langerhans cells/macrophages, but lack a phagocytic response upon infection or injury (Alemany et al., 2018; Lin et al., 2019)". Subsection “csf1rDM skin lacks highly branched Langerhans cells”. Can the authors test that the MLCs present in their mutants are actually affected in their phagocytic response? Could they perform injury assay in juvenile fish?

As described by Li et al., 2019, the lack of phagocytic capacity is inherent to MLCs. Consequently, this feature is not expected to be altered by loss of Csf1r. Furthermore, our transcriptome data confirm that metaphocytes do not express csf1r and also indicate that csf1r^DM^ metaphocytes do not differ much from the controls. In fact, the relatively few differentially expressed genes mostly belong to GO terms involved in pigment cell development and regulation of iron/ion metabolism, the relevance of which we plan to explore in follow up studies.

8) The authors conclude that "mpeg1+ cells that remain in csf1rDM zebrafish skin are, based on gene expression, morphology, cell migration and lineage tracing very likely MLCs.". The gene expression analysis is based on qPCR analysis of few selected genes on SP and DP cells. The existing molecular characterisation of the putative csf1r-independent MLCs needs being implemented by at least validating cdh1 and cldn7b, as examples of MLC genes, via qPCR and by showing the pattern of cldnh1 in the RNAseq. The pattern of such MLC markers should also be assessed in SP pMacs at earlier stages as a negative control. If possible, given the expertise shown in this paper, the authors should perform RNA-seq on FACS sorted SP (control and mutant) and DP (control) cells. MLCs are recently described cells and we still know very little about them. I think this paper, the group and others in the community would benefit in the generation of such a data set.

As mentioned above, through RNA seq analyses, we have now characterized the gene expression profiles of GFP+/DsRed- (SP) and csf1r^DM^ GFP+/DsRed- (SP) cells as well as GFP+/DsRed+ (DP) control cells. These data sets now provide an additional resource for the community to study and better understand the biology of this new cell type population.

In addition, as suggested we indicated cldnh1 in the heatmap in Figure 5. As suggested we also looked at the metaphocyte signature expressed at earlier stages as a negative control using our dataset of 28 and 50 hpf. The metaphocyte signature genes are expressed at very low levels compared to SP cells and csf1r^DM^ mpeg+ cells at juvenile stages. We included a heatmap of 28/50 hpf data showing enrichment of metaphocyte overexpressed genes in embryonic csf1r^DM^ zebrafish macrophages (Author response image 1).
